# Do Wide and Deep Networks Learn the Same Things? Uncovering How Neural Network Representations Vary with Width and Depth

**Thao Nguyen,**[*] **Maithra Raghu, & Simon Kornblith**
Google Research
{thaotn,maithra,skornblith}@google.com

## Abstract

A key factor in the success of deep neural networks is the ability to scale models to improve performance by varying the architecture depth and width. This simple property of neural network design has resulted in highly effective architectures for a variety of tasks. Nevertheless, there is limited understanding of effects of depth and width on the *learned representations*. In this paper, we study this fundamental question. We begin by investigating how varying depth and width affects model hidden representations, finding a characteristic *block structure* in the hidden representations of larger capacity (wider or deeper) models. We demonstrate that this block structure arises when model capacity is large relative to the size of the training set, and is indicative of the underlying layers preserving and propagating the dominant principal component of their representations. This discovery has important ramifications for features learned by different models, namely, representations outside the block structure are often similar across architectures with varying widths and depths, but the block structure is unique to each model. We analyze the output predictions of different model architectures, finding that even when the overall accuracy is similar, wide and deep models exhibit distinctive error patterns and variations across classes.

## 1 Introduction

Deep neural network architectures are typically tailored to available computational resources by scaling their width and/or depth. Remarkably, this simple approach to model scaling can result in state-of-the-art networks for both high- and low-resource regimes (Tan & Le, 2019). However, despite the ubiquity of varying depth and width, there is limited understanding of how varying these properties affects the final model *beyond* its performance. Investigating this fundamental question is critical, especially with the continually increasing compute resources devoted to designing and training new network architectures.

More concretely, we can ask, how do depth and width affect the final learned representations? Do these different model architectures also learn different intermediate (hidden layer) features? Are there discernible differences in the outputs? In this paper, we study these core questions, through detailed analysis of a family of ResNet models with varying depths and widths trained on CIFAR-10 (Krizhevsky et al., 2009), CIFAR-100 and ImageNet (Deng et al., 2009).

We show that depth/width variations result in distinctive characteristics in the model internal representations, with resulting consequences for representations and outputs across different model initializations and architectures. Specifically, our contributions are as follows:

- We develop a method based on centered kernel alignment (CKA) to efficiently measure the similarity of the hidden representations of wide and deep neural networks using minibatches.
- We apply this method to different network architectures, finding that representations in wide or deep models exhibit a characteristic structure, which we term the *block structure*. We study how the block structure varies across different training runs, and uncover a connection between block

---

[*]Work done as a member of the Google AI Residency program.

structure and model overparametrization — block structure primarily appears in overparameterized models.

- Through further analysis, we find that the block structure corresponds to hidden representations having a single principal component that explains the majority of the variance in the representation, which is preserved and propagated through the corresponding layers. We show that some hidden layers exhibiting the block structure can be pruned with minimal impact on performance.
- With this insight on the representational structures within a single network, we turn to comparing representations *across* different architectures, finding that models without the block structure show reasonable representation similarity in corresponding layers, but block structure representations are unique to each model.
- Finally, we look at how different depths and widths affect model outputs. We find that wide and deep models make systematically different mistakes at the level of individual examples. Specifically, on ImageNet, even when these networks achieve similar overall accuracy, wide networks perform slightly better on classes reflecting scenes, whereas deep networks are slightly more accurate on consumer goods.

## 2 RELATED WORK

Neural network models of different depth and width have been studied through the lens of universal approximation theorems (Cybenko, 1989; Hornik, 1991; Pinkus, 1999; Lu et al., 2017; Hanin & Sellke, 2017; Lin & Jegelka, 2018) and functional expressivity (Telgarsky, 2015; Raghu et al., 2017b). However, this line of work only shows that such networks can be constructed, and provides neither a guarantee of learnability nor a characterization of their performance when trained on finite datasets. Other work has studied the behavior of neural networks in the infinite width limit by relating architectures to their corresponding kernels (Matthews et al., 2018; Lee et al., 2018; Jacot et al., 2018), but substantial differences exist between behavior in this infinite width limit and the behavior of finite-width networks (Novak et al., 2018; Wei et al., 2019; Chizat et al., 2019; Lewkowycz et al., 2020). In contrast to this theoretical work, we attempt to develop empirical understanding of the behavior of practical, finite-width neural network architectures after training on real-world data.

Previous empirical work has studied the effects of width and depth upon model accuracy in the context of convolutional neural network architecture design, finding that optimal accuracy is typically achieved by balancing width and depth (Zagoruyko & Komodakis, 2016; Tan & Le, 2019). Further study of accuracy and error sets have been conducted in (Hacohen & Weinshall, 2020) (error sets over training), and (Hooker et al., 2019) (error after pruning). Other work has demonstrated that it is often possible for narrower or shallower neural networks to attain similar accuracy to larger networks when the smaller networks are trained to mimic the larger networks' predictions (Ba & Caruana, 2014; Romero et al., 2015). We instead seek to study the impact of width and depth on network internal representations and (per-example) outputs, by applying techniques for measuring similarity of neural network hidden representations (Kornblith et al., 2019; Raghu et al., 2017a; Morcos et al., 2018). These techniques have been very successful in analyzing deep learning, from properties of neural network training (Gotmare et al., 2018; Neyshabur et al., 2020), objectives (Resnick et al., 2019; Thompson et al., 2019; Hermann & Lampinen, 2020), and dynamics (Maheswaranathan et al., 2019) to revealing hidden linguistic structure in large language models (Bau et al., 2019; Kudugunta et al., 2019; Wu et al., 2019; 2020) and applications in neuroscience (Shi et al., 2019; Li et al., 2019; Merel et al., 2019; Zhang & Bellec, 2020) and medicine (Raghu et al., 2019).

## 3 EXPERIMENTAL SETUP AND BACKGROUND

Our goal is to understand the effects of depth and width on the function learned by the underlying neural network, in a setting representative of the high performance models used in practice. Reflecting this, our experimental setup consists of a family of ResNets (He et al., 2016; Zagoruyko & Komodakis, 2016) trained on standard image classification datasets CIFAR-10, CIFAR-100 and ImageNet.

For standard CIFAR ResNet architectures, the network's layers are evenly divided between three stages (feature map sizes), with numbers of channels increasing by a factor of two from one stage to the next. We adjust the network's width and depth by increasing the number of channels and layers respectively in each stage, following Zagoruyko & Komodakis (2016). For ImageNet ResNets,

ResNet-50 and ResNet-101 architectures differ only by the number of layers in the third ($14 \times 14$) stage. Thus, for experiments on ImageNet, we scale only the width or depth of layers in this stage. More details on training parameters, as well as the accuracies of all investigated models, can be found in Appendix B.

We observe that increasing depth and/or width indeed yields better-performing models. However, we will show in the following sections how they exhibit characteristic differences in internal representations and outputs, beyond their comparable accuracies.

### 3.1 Measuring Representational Similarity Using Minibatch CKA

Neural network hidden representations are challenging to analyze for several reasons including (i) their large size; (ii) their distributed nature, where important features in a layer may rely on multiple neurons; and (iii) lack of alignment between neurons in different layers. Centered kernel alignment (CKA) (Kornblith et al., 2019; Cortes et al., 2012) addresses these challenges, providing a robust way to quantitatively study neural network representations by computing the similarity between pairs of activation matrices. Specifically, we use linear CKA, which Kornblith et al. (2019) have previously validated for this purpose, and adapt it so that it can be efficiently estimated using minibatches. We describe both the conventional and minibatch estimators of CKA below.

Let $\mathbf{X} \in \mathbb{R}^{m \times p_1}$ and $\mathbf{Y} \in \mathbb{R}^{m \times p_2}$ contain representations of two layers, one with $p_1$ neurons and another $p_2$ neurons, to the same set of $m$ examples. Each element of the $m \times m$ Gram matrices $\boldsymbol{K} = \boldsymbol{X}\boldsymbol{X}^\mathsf{T}$ and $\boldsymbol{L} = \boldsymbol{Y}\boldsymbol{Y}^\mathsf{T}$ reflects the similarities between a pair of examples according to the representations contained in $\boldsymbol{X}$ or $\boldsymbol{Y}$. Let $\boldsymbol{H} = \boldsymbol{I}_n - \frac{1}{n}\mathbf{1}\mathbf{1}^\mathsf{T}$ be the centering matrix. Then $\boldsymbol{K}' = \boldsymbol{H}\boldsymbol{K}\boldsymbol{H}$ and $\boldsymbol{L}' = \boldsymbol{H}\boldsymbol{L}\boldsymbol{H}$ reflect the similarity matrices with their column and row means subtracted. HSIC measures the similarity of these centered similarity matrices by reshaping them to vectors and taking the dot product between these vectors, $\mathrm{HSIC}_0(\boldsymbol{K}, \boldsymbol{L}) = \mathrm{vec}(\boldsymbol{K}') \cdot \mathrm{vec}(\boldsymbol{L}')/(m-1)^2$. HSIC is invariant to orthogonal transformations of the representations and, by extension, to permutation of neurons, but it is not invariant to scaling of the original representations. CKA further normalizes HSIC to produce a similarity index between 0 and 1 that is invariant to isotropic scaling,

$$\mathrm{CKA}(\boldsymbol{K}, \boldsymbol{L}) = \frac{\mathrm{HSIC}_0(\boldsymbol{K}, \boldsymbol{L})}{\sqrt{\mathrm{HSIC}_0(\boldsymbol{K}, \boldsymbol{K})\mathrm{HSIC}_0(\boldsymbol{L}, \boldsymbol{L})}}. \tag{1}$$

Kornblith et al. (2019) show that, when measured between layers of architecturally identical networks trained from different random initializations, linear CKA reliably identifies architecturally corresponding layers, whereas several other proposed representational similarity measures do not. However, naive computation of linear CKA requires maintaining the activations across the entire dataset in memory, which is challenging for wide and deep networks. To reduce memory consumption, we propose to compute linear CKA by averaging HSIC scores over $k$ minibatches:

$$\mathrm{CKA}_{\mathrm{minibatch}} = \frac{\frac{1}{k}\sum_{i=1}^{k} \mathrm{HSIC}_1(\mathbf{X}_i\mathbf{X}_i^\mathsf{T}, \mathbf{Y}_i\mathbf{Y}_i^\mathsf{T})}{\sqrt{\frac{1}{k}\sum_{i=1}^{k} \mathrm{HSIC}_1(\mathbf{X}_i\mathbf{X}_i^\mathsf{T}, \mathbf{X}_i\mathbf{X}_i^\mathsf{T})}\sqrt{\frac{1}{k}\sum_{i=1}^{k} \mathrm{HSIC}_1(\mathbf{Y}_i\mathbf{Y}_i^\mathsf{T}, \mathbf{Y}_i\mathbf{Y}_i^\mathsf{T})}}, \tag{2}$$

where $\mathbf{X}_i \in \mathbb{R}^{n \times p_1}$ and $\mathbf{Y}_i \in \mathbb{R}^{n \times p_2}$ are now matrices containing activations of the $i^\mathrm{th}$ minibatch of $n$ examples sampled without replacement. In place of $\mathrm{HSIC}_0$, which is a biased estimator of HSIC, we use an unbiased estimator of HSIC (Song et al., 2012) so that the value of CKA is independent of the batch size:

$$\mathrm{HSIC}_1(\mathbf{K}, \mathbf{L}) = \frac{1}{n(n-3)}\left(\mathrm{tr}(\tilde{\mathbf{K}}\tilde{\mathbf{L}}) + \frac{\mathbf{1}^\mathsf{T}\tilde{\mathbf{K}}\mathbf{1}\mathbf{1}^\mathsf{T}\tilde{\mathbf{L}}\mathbf{1}}{(n-1)(n-2)} - \frac{2}{n-2}\mathbf{1}^\mathsf{T}\tilde{\mathbf{K}}\tilde{\mathbf{L}}\mathbf{1}\right), \tag{3}$$

where $\tilde{\mathbf{K}}$ and $\tilde{\mathbf{L}}$ are obtained by setting the diagonal entries of similarity matrices $\mathbf{K}$ and $\mathbf{L}$ to zero.

This approach of estimating HSIC based on minibatches is equivalent to the bagging block HSIC approach of Yamada et al. (2018), and converges to the same value as if the entire dataset were considered as a single minibatch, as proven in Appendix A. We use minibatches of size $n = 256$ obtained by iterating over the test dataset 10 times, sampling without replacement within each time.

## 4 Depth, Width and Model Internal Representations

We begin our study by investigating how the depth and width of a model architecture affects its internal representation structure. How do representations evolve through the hidden layers in different

architectures? How similar are different hidden layer representations to each other? To answer these questions, we use the CKA representation similarity measure outlined in Section 3.1.

We find that as networks become wider and/or deeper, their representations show a characteristic *block structure*: many (almost) consecutive hidden layers that have highly similar representations. By training with reduced dataset size, we pinpoint a connection between block structure and model overparametrization — block structure emerges in models that have large capacity relative to the training dataset.

## 4.1 Internal Representations and the Block Structure

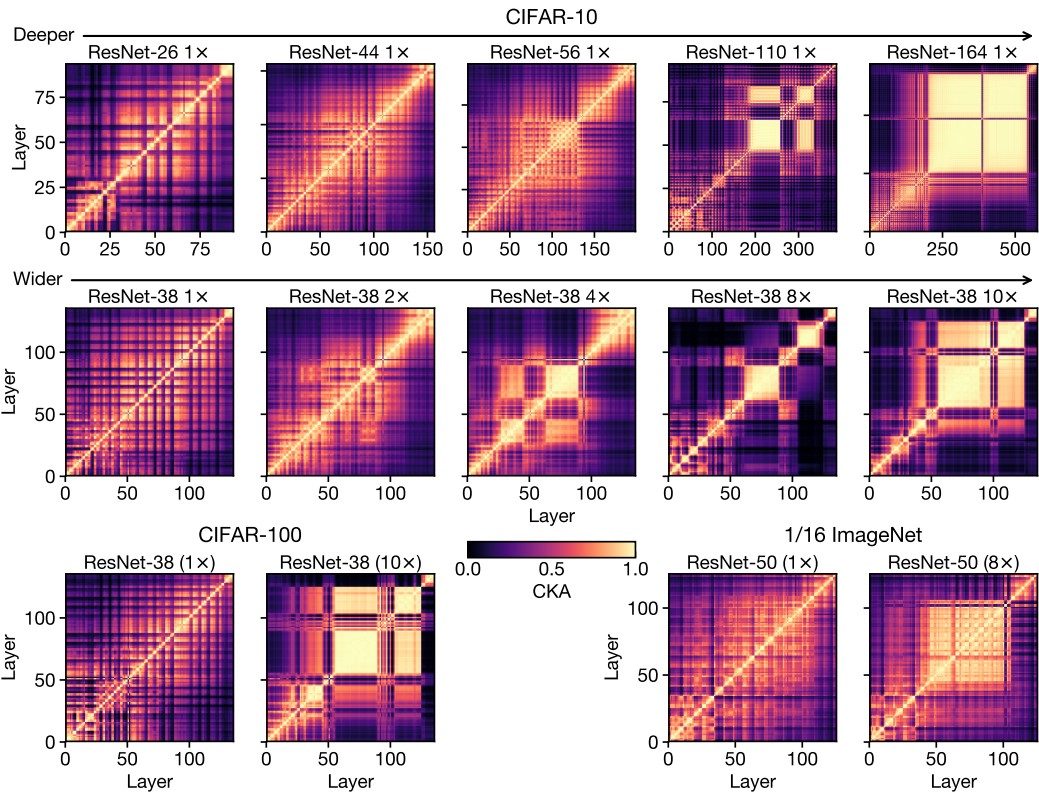

**Figure 1: Emergence of the *block structure* with increasing width or depth.** As we increase the depth or width of neural networks, we see the emergence of a large, contiguous set of layers with very similar representations — the block structure. Each of the panes of the figure computes the CKA similarity between all pairs of layers in a single neural network and plots this as a heatmap, with x and y axes indexing layers. See Appendix Figure C.1 for block structure in wide networks without residual connections.

In Figure 1, we show the results of training ResNets of varying depths (top row) and widths (bottom row) on CIFAR-10. For each ResNet, we use CKA to compute the representation similarity of all pairs of layers within the same model. Note that the total number of layers is much greater than the stated depth of the ResNet, as the latter only accounts for the convolutional layers in the network but we include *all* intermediate representations. We can visualize the result as a heatmap, with the x and y axes representing the layers of the network, going from the input layer to the output layer.

The heatmaps start off as showing a checkerboard-like representation similarity structure, which arises because representations after residual connections are more similar to other post-residual representations than representations inside ResNet blocks. As the model gets wider or deeper, we see the emergence of a distinctive *block structure* — a considerable range of hidden layers that have very high representation similarity (seen as a yellow square on the heatmap). This block structure mostly appears in the later layers (the last two stages) of the network. We observe similar results in networks without residual connections (Appendix Figure C.1).

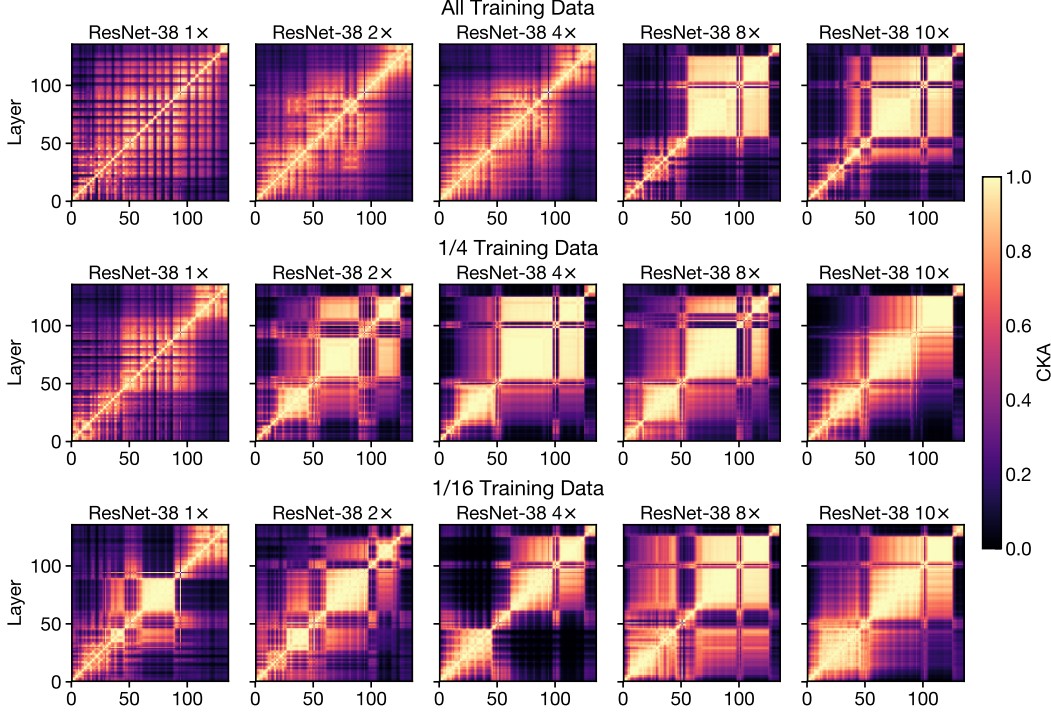

**Figure 2: Block structure emerges in narrower networks when trained on less data.** We plot CKA similarity heatmaps as we increase network width (going right along each row) and also decrease the dataset size (down each column). As a result of the increased model capacity (with respect to the task) from smaller dataset size, smaller (narrower) models now also exhibit the block structure.

**Block structure across random seeds:** In Appendix Figure D.1, we plot CKA heatmaps across multiple random seeds of a deep network and a wide network. We observe that while the exact size and position of the block structure can vary, it is present across all training runs.

## 4.2 THE BLOCK STRUCTURE AND MODEL OVERPARAMETRIZATION

Having observed that the block structure emerges as models get deeper and/or wider (Figure 1), we next study whether block structure is a result of this increase in model capacity — namely, is block structure connected to the *absolute* model size, or to the size of the model *relative* to the size of the training data?

Commonly used neural networks have many more parameters than there are examples in their training sets. However, even within this overparameterized regime, larger networks frequently achieve higher performance on held out data (Zagoruyko & Komodakis, 2016; Tan & Le, 2019). Thus, to explore the connection between *relative* model capacity and the block structure, we fix a model architecture, but *decrease* the training dataset size, which serves to inflate the relative model capacity.

The results of this experiment with varying network widths are shown in Figure 2, while the corresponding plot with varying network depths (which supports the same conclusions) can be found in Appendix Figure D.2. Each column of Figure 2 shows the internal representation structure of a fixed architecture as the amount of training data is reduced, and we can clearly see the emergence of the block structure in narrower (lower capacity) networks as less training data is used. Refer to Figures D.3 and D.4 in the Appendix for a similar set of experiments on CIFAR-100. Together, these observations indicate that block structure in the internal representations arises in models that are heavily overparameterized relative to the training dataset.

## 5 PROBING THE BLOCK STRUCTURE

In the previous section, we show that wide and/or deep neural networks exhibit a block structure in the CKA heatmaps of their internal representations, and that this block structure arises from the large

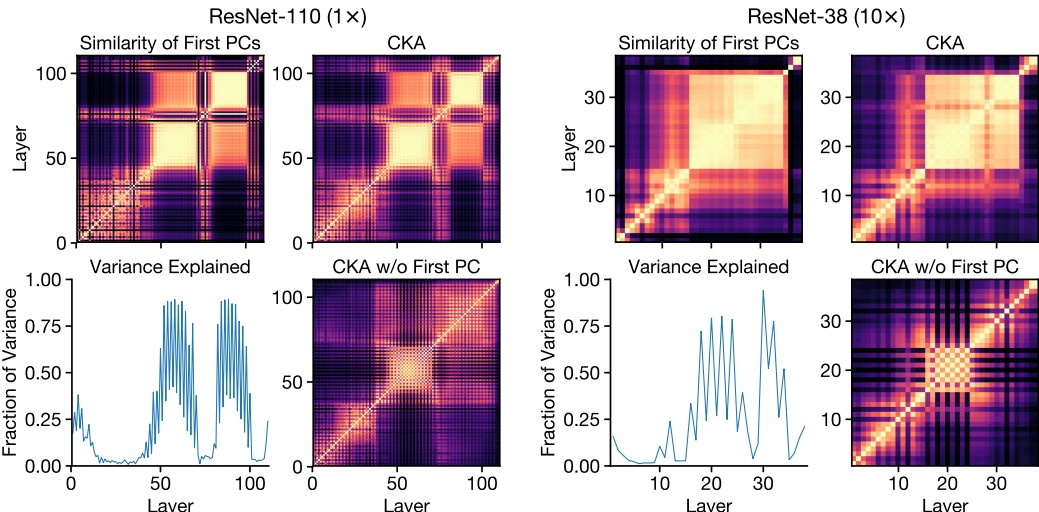

**Figure 3: Block structure arises from preserving and propagating the (dominant) first principal component of the layer representations.** Above are two sets of four plots, for layers of a deep network (left) and a wide network (right). CKA of the representations (top right), shows block structure in both networks. By comparing this to the variance explained by the top principal component of each layer representation (bottom left), we see that layers in the block structure have a highly dominant first principal component. This principal component is also preserved throughout the block structure, seen by comparing the squared cosine similarity of the first principal component across pairs of layers (top left), to the CKA representation similarity (top right). Compared to the latter, after removing the first principal component from the representations (bottom right), the block structure is highly reduced — the block structure arises from propagating the first principal component.

capacity of the models in relation to the learned task. While this latter result provides some insight into the block structure, there remains a key open question, which this section seeks to answer: what is happening to the neural network representations as they propagate through the block structure?

Through further analysis, we show that the block structure arises from the *preservation* and *propagation* of the first principal component of its constituent layer representations. Additional experiments with linear probes (Alain & Bengio, 2016) further support this conclusion and show that some layers that make up the block structure can be removed with minimal performance loss.

## 5.1 THE BLOCK STRUCTURE AND THE FIRST PRINCIPAL COMPONENT

For centered matrices of activations $\boldsymbol{X} \in \mathbb{R}^{n \times p_1}$, $\boldsymbol{Y} \in \mathbb{R}^{n \times p_2}$, linear CKA may be written as:

$$\text{CKA}(XX^{\text{T}}, YY^{\text{T}}) = \frac{\sum_{i=1}^{p_1} \sum_{j=1}^{p_2} \lambda_X^i \lambda_Y^j \langle \mathbf{u}_X^i, \mathbf{u}_Y^j \rangle^2}{\sqrt{\sum_{i=1}^{p_1} (\lambda_X^i)^2} \sqrt{\sum_{j=1}^{p_2} (\lambda_Y^j)^2}} \quad (4)$$

where $\boldsymbol{u}_X^i \in \mathbb{R}^n$ and $\boldsymbol{u}_Y^i \in \mathbb{R}^n$ are the $i^{\text{th}}$ normalized principal components of $\boldsymbol{X}$ and $\boldsymbol{Y}$ and $\lambda_X^i$ and $\lambda_Y^i$ are the corresponding squared singular values (Kornblith et al., 2019). As the fraction of the variance explained by the first principal components approaches 1, CKA reflects the squared alignment between these components $\langle \mathbf{u}_X^1, \mathbf{u}_Y^1 \rangle^2$. We find that, in networks with a visible block structure, the first principal component explains a large fraction of the variance, whereas in networks with no visible block structure, it does not (Appendix Figure D.5), suggesting that the block structure reflects the behavior of the first principal component of the representations.

Figure 3 explores this relationship between the block structure and the first principal components of the corresponding layer representations, demonstrated on a deep network (left group) and a wide network (right group). By comparing the variance explained by the first principal component (bottom left) to the location of the block structure (top right) we observe that layers belonging to the block structure have a highly dominant first principal component. Cosine similarity of the first principal components across all pairs of layers (top left) also shows a similarity structure resembling the block structure (top right), further demonstrating that the principal component is preserved throughout the block structure. Finally, removing the first principal component from the representations

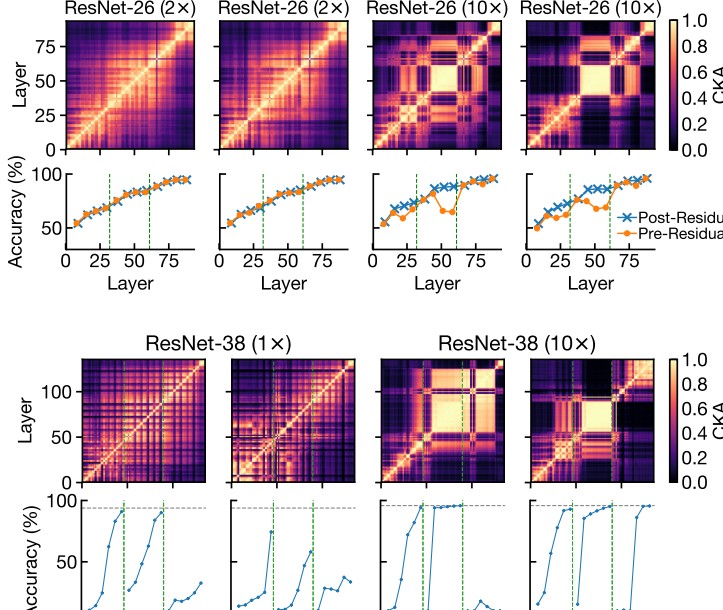

**Figure 4: Linear probe accuracy.** Top: CKA between layers of individual ResNet models, for different architectures and initializations. Bottom: Accuracy of linear probes for each of the layers before (orange) and after (blue) the residual connections.

**Figure 5: Effect of deleting blocks on accuracy for models with and without block structure.** Blue lines show the effect of deleting blocks backwards one-by-one within each ResNet stage. (Note the plateau at the block structure.) Vertical green lines reflect boundaries between ResNet stages. Horizontal gray line reflects accuracy of the full model.

nearly eliminates the block structure from the CKA heatmaps (bottom right). A full picture of how this process impacts models of increasing depth and width can be found in Appendix Figure D.6.

In contrast, for models that do not contain the block structure, we find that cosine similarity of the first principal components across all pairs of layers bears little resemblance to the representation similarity structure measured by CKA, and the fractions of variance explained by the first principal components across all layers are relatively small (see Appendix Figure D.7). Together these results demonstrate that the block structure arises from preserving and propagating the first principal component across its constituent layers.

Although layers inside the block structure have representations with high CKA and similar first principal components, each layer nonetheless computes a nonlinear transformation of its input. Appendix Figure D.8 shows that the sparsity of ReLU activations inside and outside of the block structure is similar. In particular, ReLU activations in the block structure are sometimes in the linear regime and sometimes in the saturating regime, just like activations elsewhere in the network.

## 5.2 LINEAR PROBES AND COLLAPSING THE BLOCK STRUCTURE

With the insight that the block structure is preserving key components of the representations, we next investigate how these preserved representations impact task performance throughout the network, and whether the block structure can be collapsed in a way that minimally affects performance.

In Figure 4, we train a linear probe (Alain & Bengio, 2016) for each layer of the network, which maps from the layer representation to the output classes. In models without the block structure (first 2 panes), we see a monotonic increase in accuracy throughout the network, but in models with the block structure (last 2 panes), linear probe accuracy shows little improvement inside the block structure. Comparing the accuracies of probes for layers pre- and post-residual connections, we find that these connections play an important role in preserving representations in the block structure.

Informed by these results, we proceed to pruning blocks one-by-one from the end of each residual stage, while keeping the residual connections intact, and find that there is little impact on test accuracy when blocks are dropped from the middle stage (Figure 5), unlike what happens in models without block structure. When compared across different seeds, the magnitude of the drop in accuracy appears to be connected to the size and the clarity of the block structure present. This result suggests that block structure could be an indication of redundant modules in model design, and that the similarity of its constituent layer representations could be leveraged for model compression.

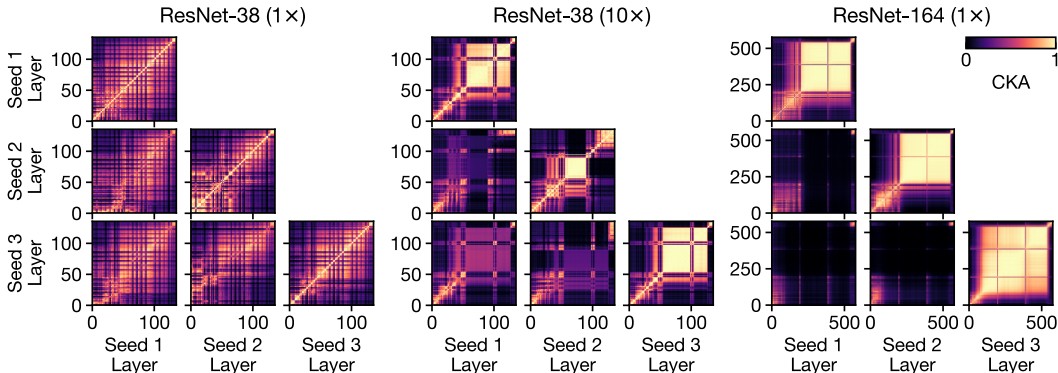

**Figure 6: Representations within "block structure" differ across initializations.** Each group of plots shows CKA between layers of models with the same architecture but different initializations (off the diagonal) or within a single model (on the diagonal). For narrow, shallow models such as ResNet-38 (1×), there is no block structure, and CKA across initializations closely resembles CKA within a single model. For wider (middle) and deeper (right) models, representations within the block structure are unique to each model.

## 6 Depth and Width Effects on Representations Across Models

The results of the previous sections help characterize effects of varying depth and width on a (single) model's internal representations, specifically, the emergence of the block structure with increased capacity, and its impacts on how representations are propagated through the network. With these insights, we next look at how depth and width affect the hidden representations *across* models. Concretely, are learned representations similar across models of different architectures and different random initializations? How is this affected as model capacity is changed?

We begin by studying the variations in representations across different training runs of the same model architecture. Figure 6 illustrates CKA heatmaps for a smaller model (left), wide model (middle) and deep model (right), trained from random initializations. The smaller model does not have the block structure, and representations across seeds (off diagonal plots) exhibit the same grid-like similarity structure as within a single model. The wide and deep models show block structure in all their seeds (as seen in plots along the diagonal), and comparisons across seeds (off-diagonal plots) show that while layers not in the block structure exhibit some similarity, the block structure representations are highly dissimilar across models.

Appendix Figure E.1 shows results of comparing CKA *across* different architectures, controlled for accuracy. Wide and deep models without the block structure do exhibit representation similarity with each other, with corresponding layers broadly being of the same *proportional* depth in the model. However, similar to what we observe in Figure 6, the block structure representations remain unique to each model.

## 7 Depth, Width and Effects on Model Predictions

To conclude our investigation on the effects of depth and width, we turn to understanding how the characteristic properties of internal representations discussed in the previous sections influence the outputs of the model. How diverse are the predictions of different architectures? Are there examples that wide networks are more likely to do well on compared to deep networks, and vice versa?

By training populations of networks on CIFAR-10 and ImageNet, we find that there is considerable diversity in output predictions at the *individual example* level, and broadly, architectures that are more similar in structure have more similar output predictions. On ImageNet we also find that there are statistically significant differences in class-level error rates between wide and deep models, with the former exhibiting a small advantage in identifying classes corresponding to scenes over objects.

Figure 7a compares per-example accuracy for groups of 100 architecturally identical deep models (ResNet-62) and wide models (ResNet-14 (2×)), all trained from different random initializations on CIFAR-10. Although the *average* accuracy of these groups is statistically indistinguishable, they tend to make different errors, and differences between groups are substantially larger than expected

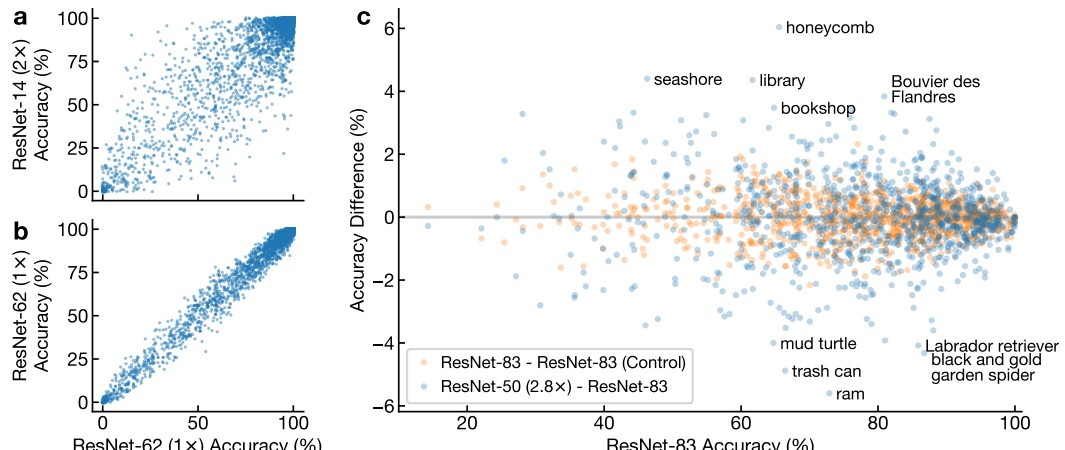

**Figure 7: Systematic per-example and per-class performance differences between wide and deep models.**
**a**: Comparison of accuracy on individual examples for 100 ResNet-62 (1×) and ResNet-14 (2×) models, which have statistically indistinguishable accuracy on the CIFAR-10 test set. **b**: Same as (a), for disjoint sets of 100 architecturally identical ResNet-62 models trained from different initializations. See Figure F.1 for a similar plot for ResNet-14 (2×) models. **c**: Accuracy differences on ImageNet classes for ResNets between models with increased width (y-axis) or depth (x-axis) in the third stage. Orange dots reflect difference between two sets of 50 architecturally identical deep models (i.e., different random initializations of ResNet-83).

by chance (Figure 7b). Examples of images with large accuracy differences are shown in Appendix Figure F.1, while Appendix Figures F.2 and F.3 further explore patterns of example accuracy for networks of different depths and widths, respectively. As the architecture becomes wider or deeper, accuracy on many examples increases, and the effect is most pronounced for examples where smaller networks were often but not always correct. At the same time, there are examples that larger networks are *less* likely to get right than smaller networks. We show similar results for ImageNet networks in Appendix Figure F.4.

We next ask whether wide and deep ImageNet models have systematic differences in accuracy at the class level. As shown in Figure 7c, there are small but statistically significant differences in accuracy for $419/1000$ classes ($p < 0.05$, Welch's $t$-test), accounting for 11% of the variance in the differences in example-level accuracy (see Appendix F.3). Three of the top 5 classes that are more likely to be correctly classified by wide models reflect scenes rather than objects (seashore, library, bookshop). Indeed, the wide architecture is significantly more accurate on the 68 ImageNet classes descending from "structure" or "geological formation" ($74.9\% \pm 0.05$ vs. $74.6\% \pm 0.06$, $p = 6 \times 10^{-5}$, Welch's t-test). Looking at synsets containing $> 50$ ImageNet classes, the deep architecture is significantly more accurate on the 62 classes descending from "consumer goods" ($72.4\% \pm 0.07$ vs. $72.1\% \pm 0.06$, $p = 0.001$; Table F.2). In other parts of the hierarchy, differences are smaller; for instance, both models achieve 81.6% accuracy on the 118 dog classes ($p = 0.48$).

## 8  CONCLUSION

In this work, we study the effects of width and depth on neural network representations. Through experiments on CIFAR-10, CIFAR-100 and ImageNet, we have demonstrated that as either width or depth increases relative to the size of the dataset, analysis of hidden representations reveals the emergence of a characteristic *block structure* that reflects the similarity of a dominant first principal component, propagated across many network hidden layers. Further analysis finds that while the block structure is unique to each model, other learned features are shared across different initializations and architectures, particularly across relative depths of the network. Despite these similarities in representational properties and performance of wide and deep networks, we nonetheless observe that width and depth have different effects on network predictions at the example and class levels. There remain interesting open questions on how the block structure arises through training, and using the insights on network depth and width to inform optimal task-specific model design.

ACKNOWLEDGEMENTS

We thank Gamaleldin Elsayed for helpful feedback on the manuscript.

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

# Appendix

## A CONVERGENCE OF MINIBATCH HSIC

**Proposition 1.** *Let $\boldsymbol{K} \in \mathbb{R}^{m \times m}$ and $\boldsymbol{L} \in \mathbb{R}^{m \times m}$ be two kernel matrices constructed by applying kernel functions $k$ and $l$ respectively to all pairs of examples in a dataset $\mathcal{D}$. Form $c$ random partitionings $p$ of $\mathcal{D}$ into $m/n$ minibatches $b$ of size $n$, and let $\tilde{\boldsymbol{K}}^{b,p} \in \mathbb{R}^{n \times n}$ and $\tilde{\boldsymbol{L}}^{b,p} \in \mathbb{R}^{n \times n}$ be kernel matrices constructed by applying kernels $k$ and $l$ to all pairs of examples within each minibatch. Define $U_0 = \mathrm{HSIC}_1(\boldsymbol{K}, \boldsymbol{L})$, the value of $\mathrm{HSIC}_1$ applied to the full dataset, and $\tilde{U}_p = \frac{n}{m} \sum_{b=1}^{m/n} \mathrm{HSIC}_1(\boldsymbol{K}^{b,p}, \boldsymbol{L}^{b,p})$, the average value of $\mathrm{HSIC}_1$ over the minibatches in partitioning (epoch) $p$. Then $\frac{1}{c} \sum_{p=1}^{c} \tilde{U}_p \xrightarrow{P} U_0$ as $c \to \infty$.*

*Proof.* Let $\boldsymbol{i}_4^m$ be the set of all 4-tuples of indices between 1 and m where each index occurs exactly once. As proven in Theorem 3 of Song et al. (2012), $U_0$ is a U-statistic:

$$U_0 = \mathrm{HSIC}_1(\boldsymbol{K}, \boldsymbol{L}) = \frac{(m-4)!}{m!} \sum_{S \in \boldsymbol{i}_4^m} h(K_S, L_S), \tag{5}$$

where $K_{(i,j,q,r)} = (K_{i,j}, K_{i,q}, K_{i,r}, K_{j,q}, K_{j,r}, K_{q,r})$ and the kernel of the U-statistic $h$ is defined in Song et al. (2012). Let $\delta_S^b$ be 1 if the 4-tuple of dataset indices $S$ is selected in minibatch $b$ and 0 otherwise. Then:

$$\tilde{U}_p = \frac{(n-4)!}{n!} \frac{n}{m} \sum_{b=1}^{m/n} \sum_{S \in \boldsymbol{i}_4^m} \delta_S^b h(K_S, L_S). \tag{6}$$

Taking the expectation with respect to $\boldsymbol{\delta}$, and noting that $\boldsymbol{\delta}$ is independent of $h(K_S, L_S)$,

$$\mathbb{E}_{\boldsymbol{\delta}}[\tilde{U}_p] = \frac{(n-4)!}{n!} \frac{n}{m} \sum_{b=1}^{m/n} \sum_{S \in \boldsymbol{i}_4^m} \mathbb{E}_{\boldsymbol{\delta}} \left[ \delta_S^b h(K_S, L_S) \right] \tag{7}$$

$$= \frac{(n-4)!}{n!} \frac{n}{m} \sum_{b=1}^{m/n} \sum_{S \in \boldsymbol{i}_4^m} \mathbb{E}_{\boldsymbol{\delta}} \left[ \delta_S^b \right] h(K_S, L_S). \tag{8}$$

By symmetry, $\mathbb{E}_{\boldsymbol{\delta}} \left[ \delta_S^b \right]$ is the same for all example and batch indices. Specifically, there are $n!/(n-4)!$ 4-tuples that can be formed from each batch and $m!/(m-4)!$ 4-tuples that can be formed from the entire dataset, so the probability that a given 4-tuple is in a given batch is $\mathbb{E}_{\boldsymbol{\delta}} \left[ \delta_S^b \right] = (n!/(n-4)!)/(m!/(m-4)!)$. Thus:

$$\mathbb{E}_{\boldsymbol{\delta}}[\tilde{U}_p] = \frac{(m-4)!}{m!} \sum_{S \in \boldsymbol{i}_4^m} h(K_S, L_S) = U_0. \tag{9}$$

The minibatch indicators $\delta_S^b$ are either 0 or 1, so their variances and covariances are bounded, and the weighted sum in Eq. 6 has finite variance. Thus, by the law of large numbers, $\frac{1}{c} \sum_{p=1}^{c} \tilde{U}_p \xrightarrow{P} U_0$ as $p \to \infty$. $\qquad \square$

## B TRAINING DETAILS

Our CIFAR-10 and CIFAR-100 networks follow the same architecture as He et al. (2016); Zagoruyko & Komodakis (2016). We train a set of models where we fix the width multiplier of deep networks to 1 and experiment with models of depths 32, 44, 56, 110, 164. On CIFAR-100, the block structure only appears at a greater depth so we also include depths 218 and 224 in our investigation. For wide networks, we examine width multipliers of 1, 2, 4, 8 and 10 and depths of 14, 20, 26, and 38. We use SGD with momentum of 0.9, together with a cosine decay learning rate schedule and batch size of 128, to train each model for 300 epochs. Models are trained with standard CIFAR-10 data augmentation comprising random flips and translations of up to 4 pixels.

Each depth and width configuration is trained with 10 different seeds for CKA analysis, and 200 seeds for model predictions comparison.

On ImageNet, we start with the ResNet-50 architecture and increase depth or width in the third stage only, following the scaling approach of (He et al., 2016). We train for 120 epochs using SGD with momentum of 0.9 and a cosine decay learning rate schedule at a batch size of 256. We use 100 seeds for model prediction comparison.

For experiments with reduced dataset size, we subsample the training data from the original CIFAR training set by the corresponding proportion, keeping the number of samples for each class the same. All CKA results are then computed based on the full CIFAR test set.

**Table B.1: Accuracy of examined neural networks on CIFAR-10 and CIFAR-100.**

| Depth | Width | CIFAR-10 Test Accuracy (%) | CIFAR-100 Test Accuracy (%) |
|-------|-------|-----------------------------|------------------------------|
| 32 | 1 | 93.5 | 71.2 |
| 44 | 1 | 94.0 | 72.0 |
| 56 | 1 | 94.2 | 73.3 |
| 110 | 1 | 94.3 | 74.0 |
| 164 | 1 | 94.4 | 73.9 |
| 14 | 1 | 92.0 | 67.8 |
| 14 | 2 | 94.1 | 72.9 |
| 14 | 4 | 95.4 | 77.0 |
| 14 | 8 | 95.9 | 80.0 |
| 14 | 10 | 96.0 | 80.2 |
| 20 | 1 | 92.8 | 69.4 |
| 20 | 2 | 94.6 | 74.4 |
| 20 | 4 | 95.4 | 77.6 |
| 20 | 8 | 96.0 | 80.2 |
| 20 | 10 | 95.8 | 80.8 |
| 26 | 1 | 93.3 | 70.5 |
| 26 | 2 | 94.9 | 75.8 |
| 26 | 4 | 95.6 | 79.3 |
| 26 | 8 | 95.9 | 80.9 |
| 26 | 10 | 95.8 | 81.0 |
| 38 | 1 | 93.8 | 72.3 |
| 38 | 2 | 95.1 | 75.9 |
| 38 | 4 | 95.5 | 78.6 |
| 38 | 8 | 95.7 | 79.8 |
| 38 | 10 | 95.7 | 80.5 |

## C    BLOCK STRUCTURE IN A DIFFERENT ARCHITECTURE

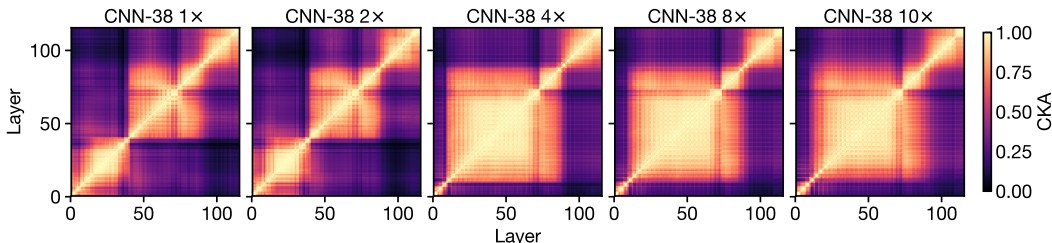

**Figure C.1: Block structure also appears in models without residual connections.** We remove residual connections from existing CIFAR-10 ResNets and plot CKA heatmaps for layers in the resulting architecture after training. Since the lack of residual connections prevents deep networks from performing well on the task, here we only show the representational similarity for models of increasing width. As previously observed in Figure 1, the block structure emerges in higher capacity models.

## D    PROBING THE BLOCK STRUCTURE

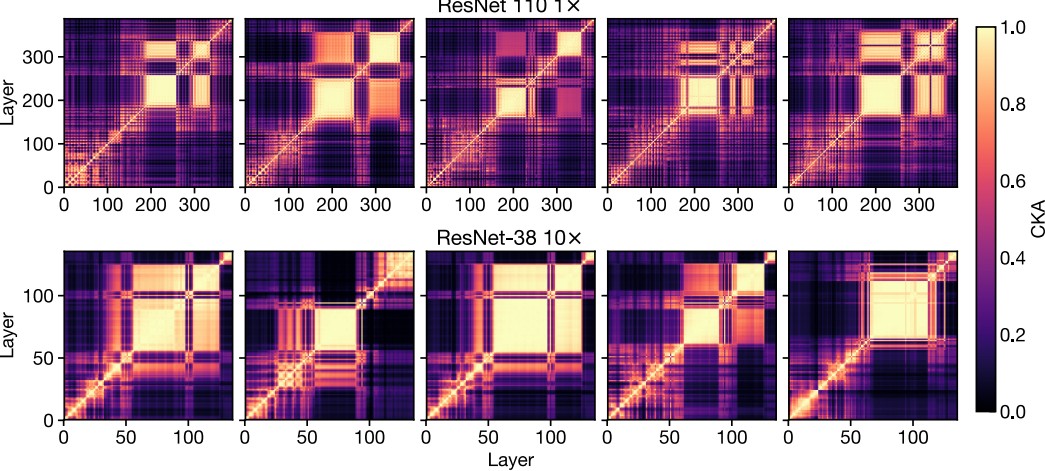

**Figure D.1: Block structure varies across random initializations.** We plot CKA heatmaps as in Figure 1 for 5 random seeds of a deep model (top row) and a wide model (bottom row) trained on CIFAR-10. While the size and position vary, the block structure is clearly visible in all seeds.

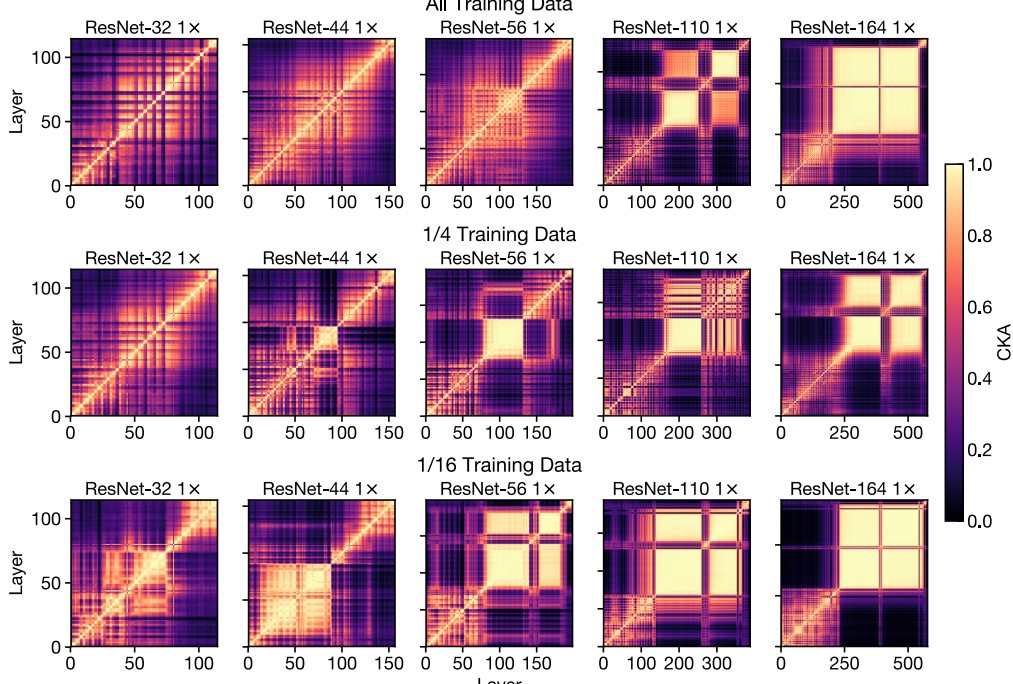

**Figure D.2: Block structure emerges in shallower networks when trained on less data (CIFAR-10).** We plot CKA similarity heatmaps as we increase network depth (going right along each row) and also decrease the size (down each column) of training data. Similar to the observation made in Figure 2, as a result of the increased model capacity (with respect to the task) from smaller dataset size, smaller (shallower) models now also exhibit the block structure.

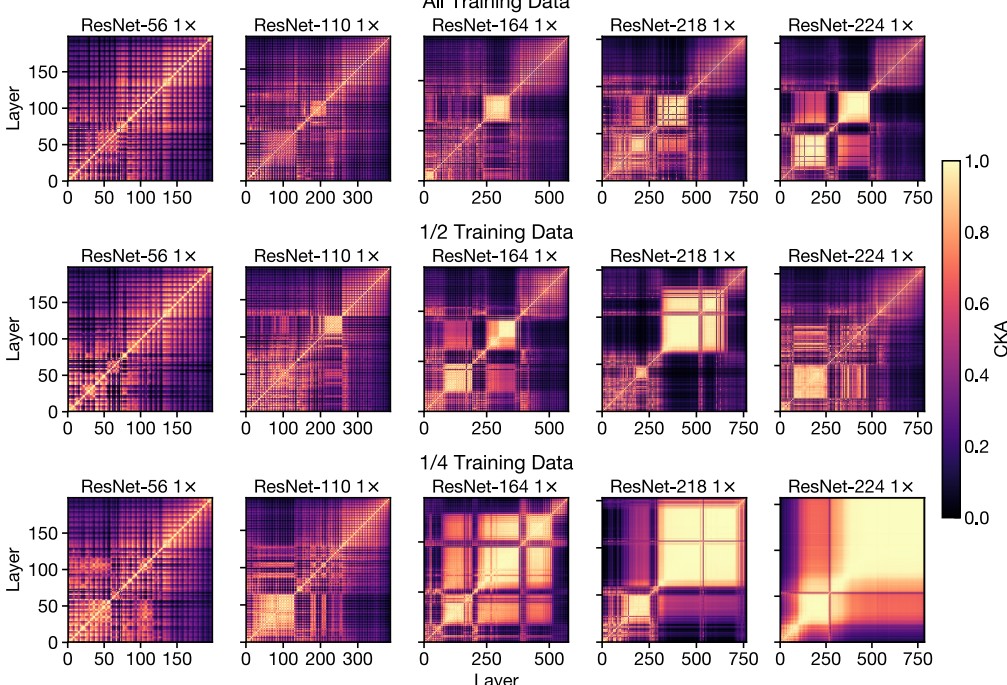

**Figure D.3: Block structure emerges in shallower networks when trained on less data (CIFAR-100).** We plot CKA similarity heatmaps as we increase network depth (going right along each row) and also decrease the size of training data (down each column). Similar to the observation made in Figure 2, as a result of the increased model capacity (with respect to the task) from smaller dataset size, smaller (shallower) models now also exhibit the block structure.

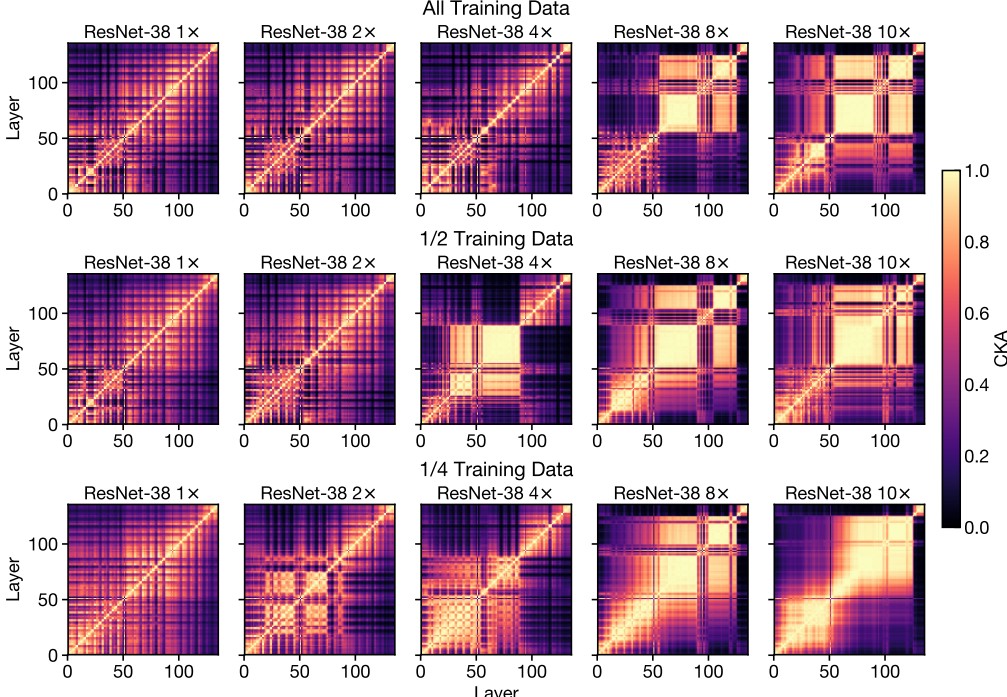

**Figure D.4: Block structure emerges in narrower networks when trained on less data (CIFAR-100).** We plot CKA similarity heatmaps as we increase network width (going right along each row) and also decrease the size (down each column) of training data. Similar to the observation made in Figure 2, as a result of the increased model capacity (with respect to the task) from smaller dataset size, smaller (narrower) models now also exhibit the block structure.

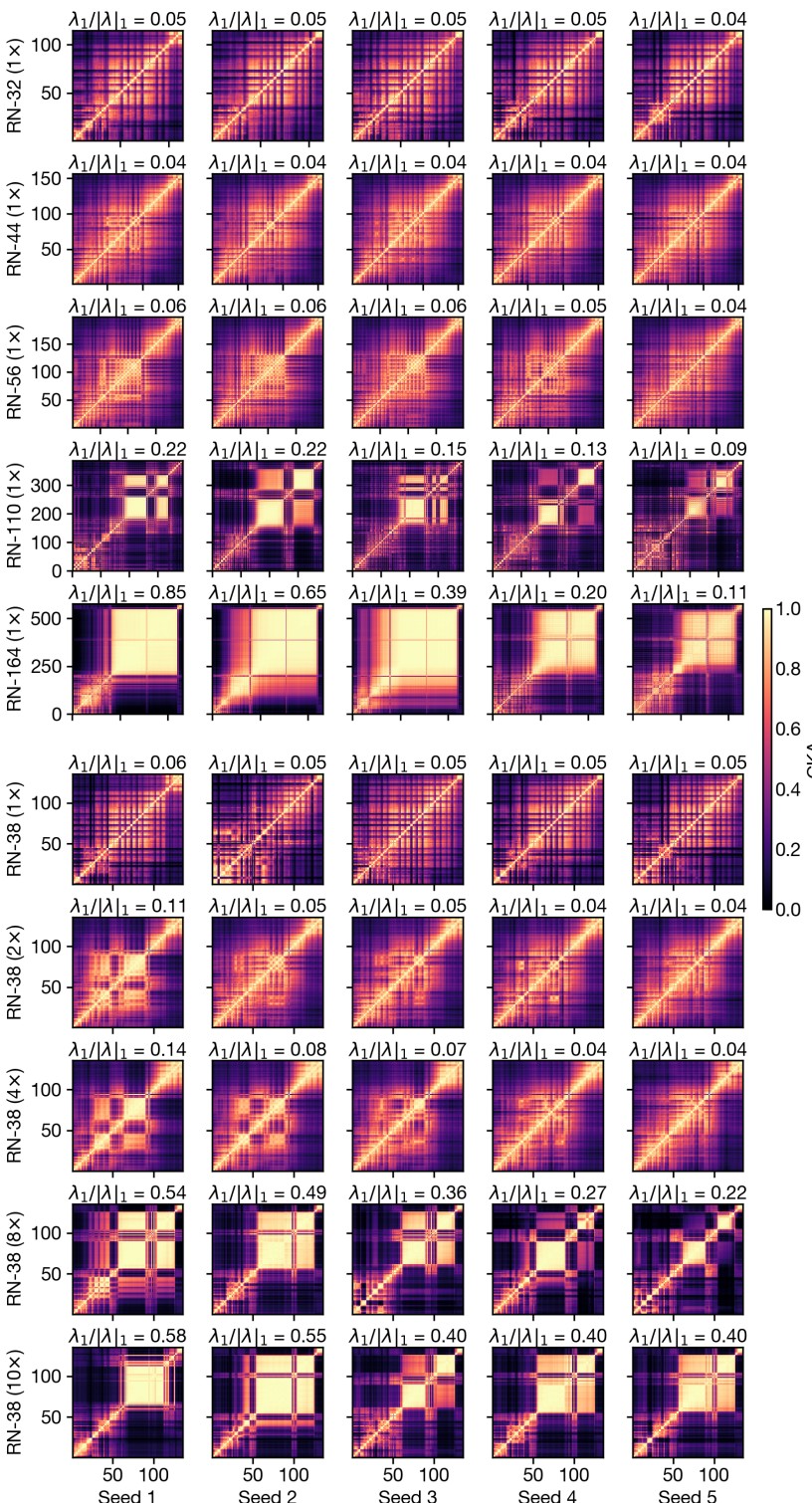

**Figure D.5: Top principal component explains a large fraction of variance in the activations of models with block structure.** Each row shows a different model configuration that is trained on CIFAR-10, with the first 5 rows showing models of increasing depth, and the last 5 rows models of increasing width. Columns correspond to different seeds. Each heatmap is labeled with the fraction of variance explained by the top principal component of activations combined from the last 2 stages of the model (where block structure is often found). Rows (seeds belonging to the same architecture) are sorted by decreasing value of the proportion of variance explained. We observe that this variance measure is significantly higher in model seeds where the block structure is present.

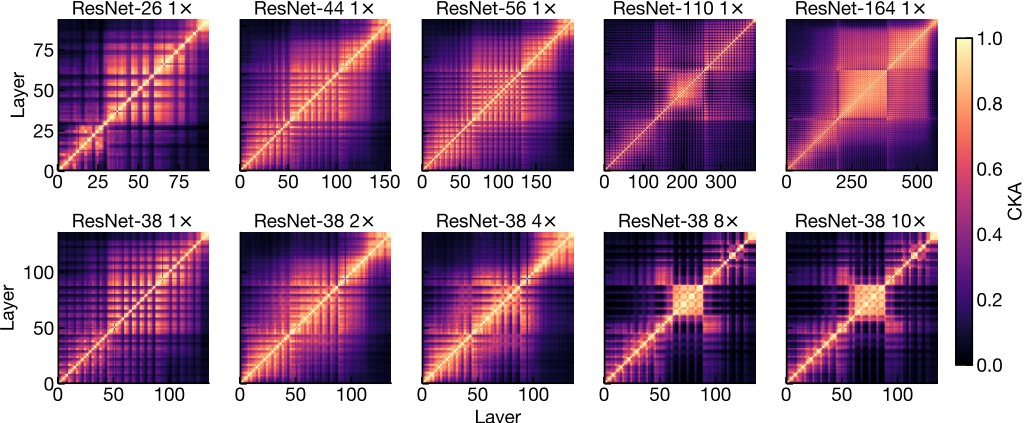

**Figure D.6: How the representational structure evolves with increasing depth and width when the first principal component is removed.** We plot CKA similarity heatmaps as models become deeper (top row) and wider (bottom row), with the top principal component of their internal representations removed. Compared to Figure 1, we observe that while this process significantly eliminates the block structure in large capacity models (as also shown in Figure 3), it has negligible impact on the representational structures of smaller models (where no block structure is present). The latter is not surprising, since the first principal component doesn't account for a large fraction of the variance in representations of these models, as demonstrated in Appendix Figure D.5 above.

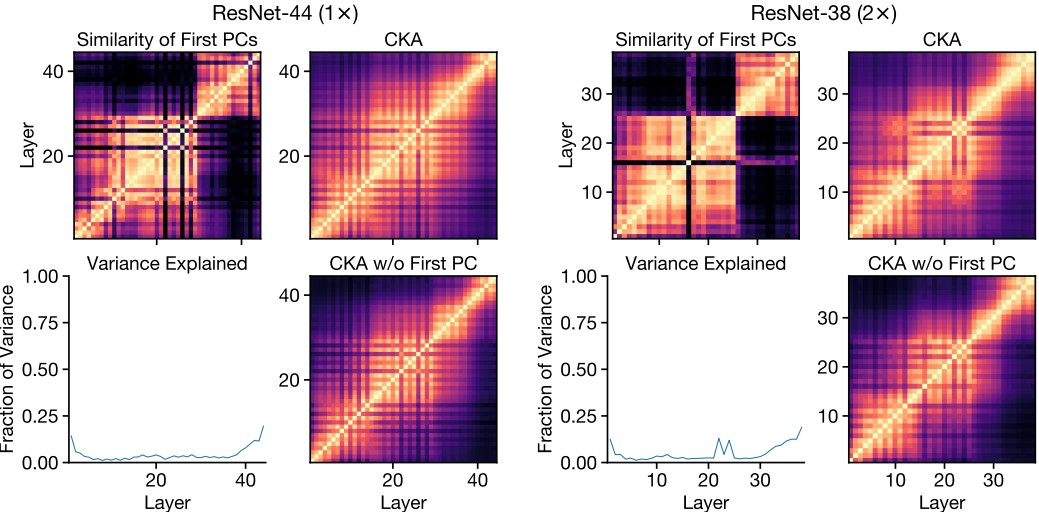

**Figure D.7: Relationship between the representation similarity structure and the first principal component in networks without block structure.** Above are two sets of four plots, for layers of a deep network (left) and a wide network (right), that don't contain block structure. In contrast to Figure 3, the first principal component of each layer representation only accounts for a small fraction of variance in the representation (bottom left). Comparing the squared cosine similarity of the first principal component across pairs of layers (top left), to the CKA representation similarity (top right), we find that these two structures don't resemble each other. Last but not least, the representational structure of each model remains mostly unchanged after the first principal component is removed (bottom right).

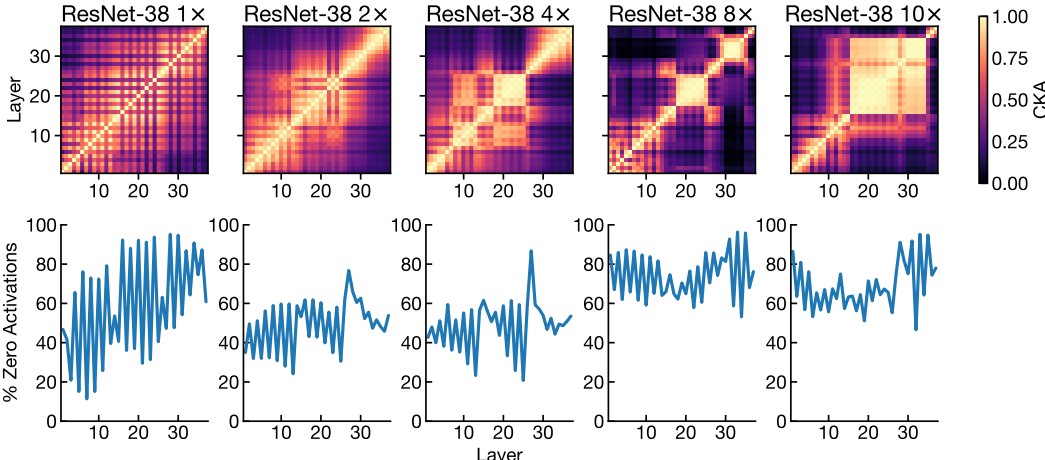

**Figure D.8: ReLU activations inside and outside the block structure are similarly sparse.** To rule out the possibility that the block structure arises because layers inside it behave linearly, we measured the sparsity of the ReLU activations. We observe that a significant proportion of activations are always non-zero, and the proportion is similar inside and outside the block structure. Thus, although layers inside the block structure have similar representations, each layer still applies a nonlinear transformation to its input.

# E    REPRESENTATIONS ACROSS MODELS

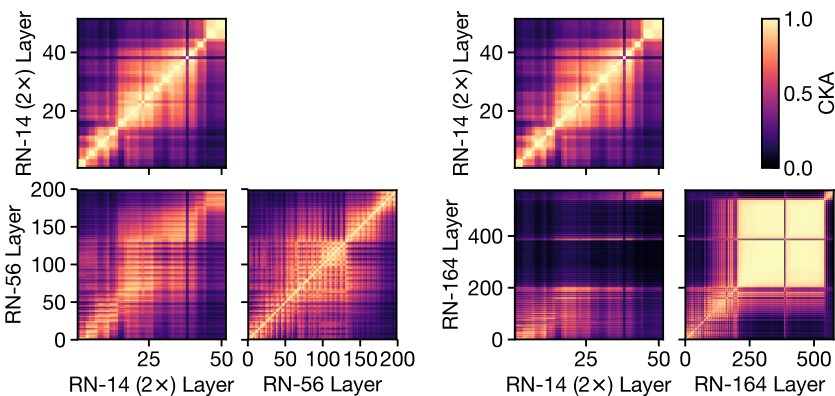

**Figure E.1: Representations align between models of different widths and depths when no block structure is present.** In each group of heatmaps, top left and bottom right show CKA within a single model trained on CIFAR-10. Bottom left shows CKA for all pairs of layers between these (non-architecturally-identical) models, which have similar test performance. In the absence of block structure (left group), representations at the same relative depths are similar across models. But when comparison involves models with block structure (right group), representations within the block structure are dissimilar to those of the other model.

# F    EXAMPLE- AND CLASS-LEVEL ACCURACY DIFFERENCES

## F.1    EFFECT OF VARYING WIDTH AND DEPTH ON CIFAR-10 PREDICTIONS

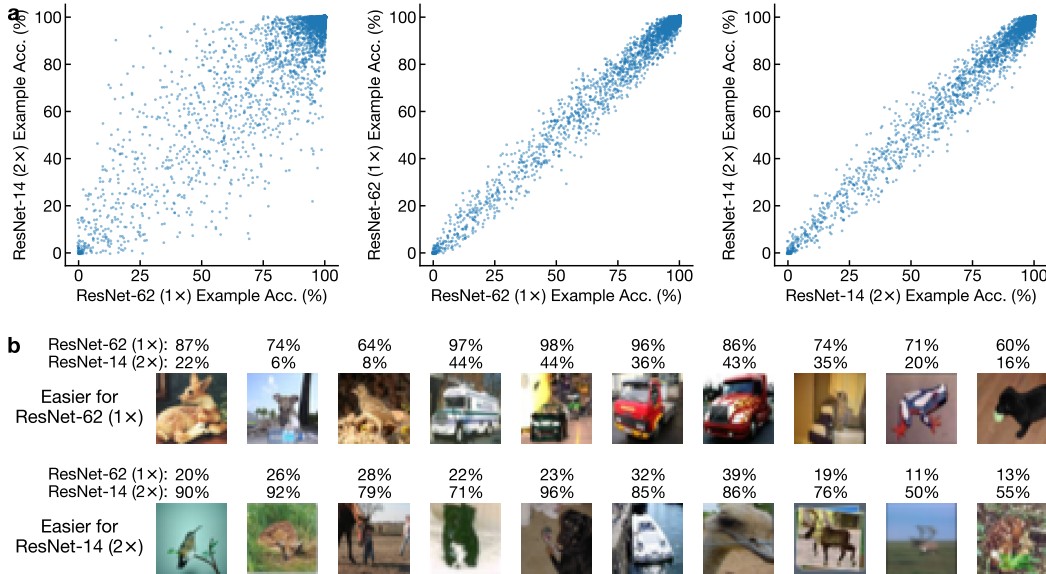

**Figure F.1: Systematic per-example performance differences between wide and deep models on CIFAR-10.** Comparison of predictions of 200 ResNet-62 ($1\times$) and ResNet-14 ($2\times$) models, which have statistically indistinguishable accuracy on the CIFAR-10 test set (mean $\pm$ SEM $94.09 \pm 0.01$ vs. $94.08 \pm 0.01$, $t(199) = 0.73$, $p = 0.47$). **a** Scatter plots of per-example accuracy for 100 ResNet-14 ($2\times$) models vs. 100 ResNet-62 ($1\times$) models (left) show substantially higher dispersion than corresponding plots for disjoint sets of 100 architecturally identical models trained from different initializations (middle and right). **b**: Examples with the highest accuracy differences between the two types of models. Accuracies are reported on a subset of models that is disjoint from those used to select the examples.

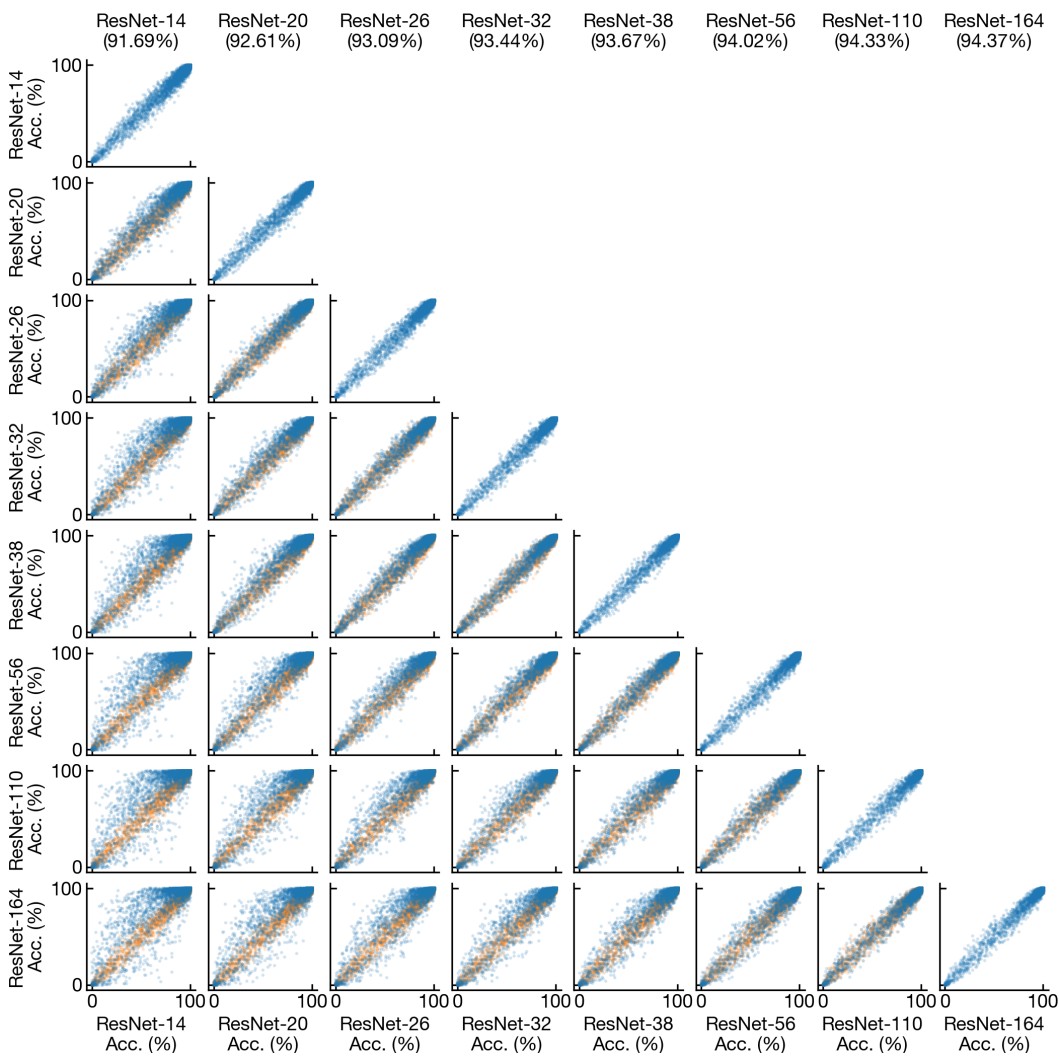

**Figure F.2: Effect of depth on example accuracy.** Scatter plots of per-example accuracies of ResNet models with different depths on CIFAR-10. Blue dots indicate per-example accuracies of two groups of 100 networks each with different architectures indicated by axes labels. Orange dots show the distribution for groups of architecturally identical models, copied from the plot on the diagonal above. Accuracy of each model is shown at the top.

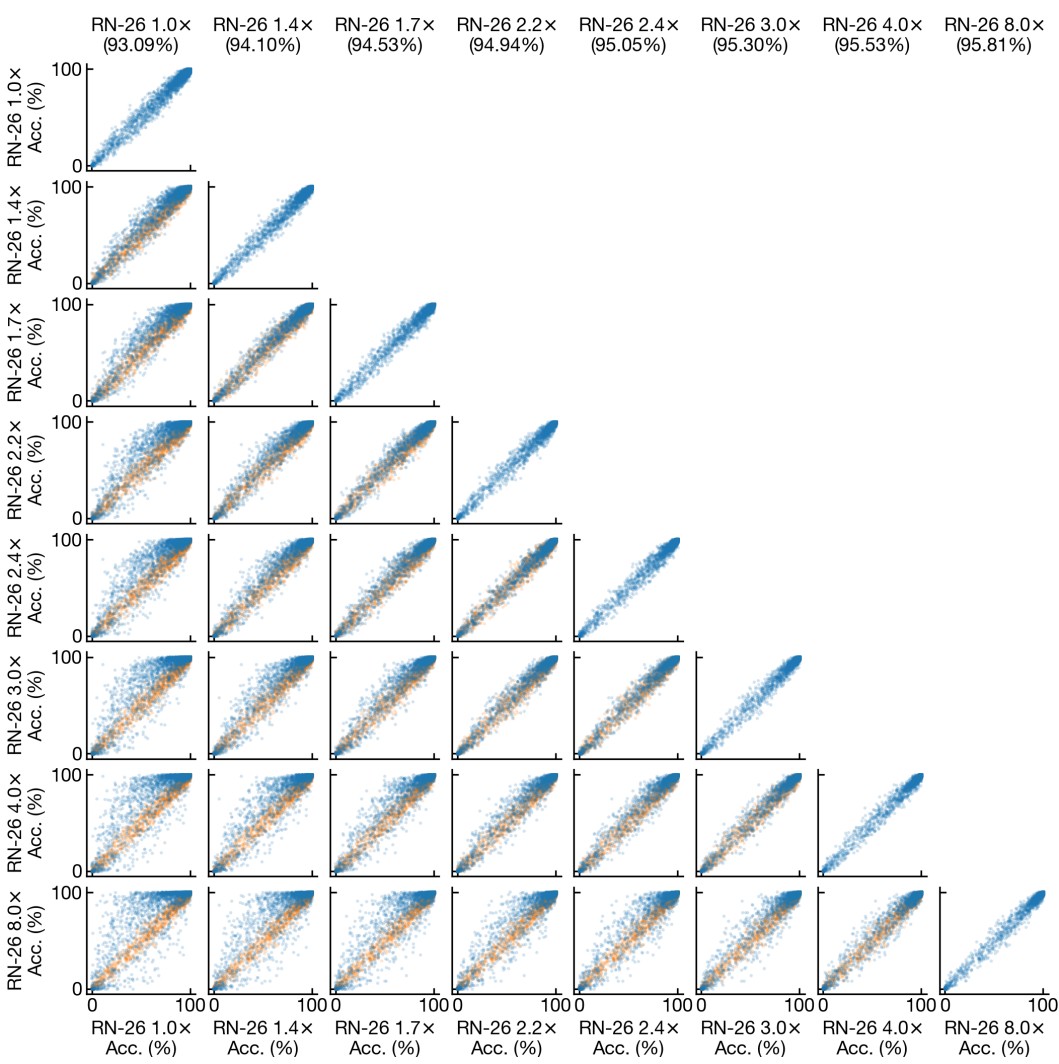

**Figure F.3: Effect of width on example accuracy.** Scatter plots of per-example accuracies of ResNet models with different widths on CIFAR-10. Blue dots indicate per-example accuracies of two groups of 100 networks each with different architectures indicated by axes labels. Orange dots show the expected distribution for groups of architecturally identical models, copied from the plot on the diagonal above. Accuracy of each model is shown at the top.

## F.2 EFFECT OF VARYING WIDTH AND DEPTH ON IMAGENET PREDICTIONS

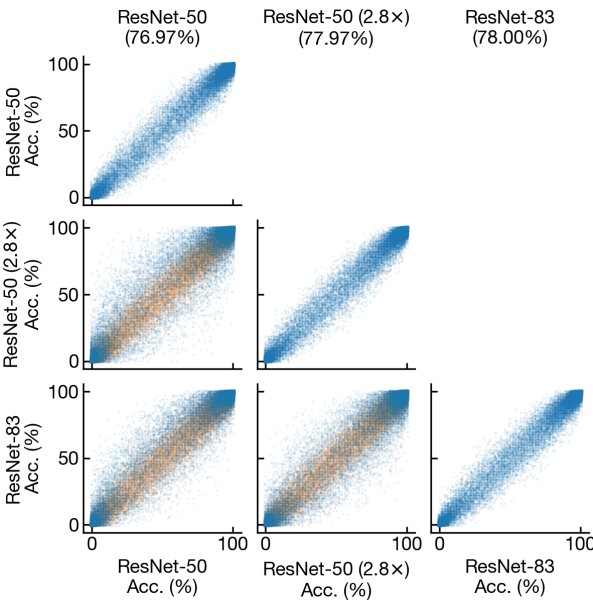

**Figure F.4: Systematic per-example performance differences between wide and deep models on ImageNet.** Scatter plots of per-example accuracy averaged across 50 vanilla ResNet-50 ($1\times$) models versus that for groups of 50 models with increased depth ($6 \rightarrow 17$ blocks, "ResNet-83") or width ($2.8\times$ wider) in the 3rd stage. Orange dots in plots show the expected distribution for two groups of 50 architecturally identical models, copied from the plot on the diagonal above. The deeper and wider models have very similar but statistically distinguishable accuracy (mean $\pm$ SEM for deeper model: $78.00 \pm 0.01$, wider model: $77.97 \pm 0.01$, $t(99) = 2.0, p = 0.047$).

## F.3 EFFECT SIZES FOR CLASS-LEVEL EFFECTS

We measure how much of the difference between the example-level predictions of the wide and deep ImageNet ResNets in Section 7 could be explained by the classes to which they belonged by fitting a set of three models. Model A attempts to model whether each prediction was correct or incorrect as a linear combination factors corresponding to the example ID and whether the prediction came from a wide or deep model (`statsmodels` formula `y ~ C(example_id) + C(wide_or_deep)`). Model B includes a factor corresponding to the example ID as well as factors corresponding to the interaction between the class ID and the type of model the prediction came from (`statsmodels` formula `y ~ C(example_id) + C(wide_or_deep) * C(class_id)`). Model C includes the interaction between the example ID and the type of model, and corresponds to simply measuring the average accuracy separately for both types of models (`statsmodels` formula `y ~ C(example_id) * C(wide_or_deep)`). Note that model C is nested inside model B, which is nested inside model A.

We seek to measure how much of the variability in predictions that can be explained by model C but not model A can be explained by model B. We fit these models with logistic regression and measure the residual variance $\text{Var}_Q = \sum_{i=1}^n (y_i - \pi_i)^2/n$, where $n$ is the number of examples $\times$ the number of models, $y_i$ is either 1 or 0 depending on whether the CNN's prediction was correct or incorrect, and $\pi_i$ is the output probability from logistic regression model $Q$. We then compute the squared differences between $y$ and the predictions of the logistic regression model:

$$v^2 = \frac{\text{Var}_A - \text{Var}_B}{\text{Var}_A - \text{Var}_C} = 0.11. \tag{10}$$

This approach is analogous to the pseudo-$R^2$ of Efron (1978).

Finally, we can compare the AIC values of the logistic regression models, shown in the table below. Because the models are nested GLMs, we can also test for statistical significance using a $\chi^2$ test, which is highly significant for each pair of nested models. We do not report p-values because they are 0 to within machine precision.

**Table F.1: AIC for models A, B, and C, described above**

| Model | AIC |
|---|---|
| A: Example + Model | 3011367 |
| B: Example + Class * Model | 3006889 |
| C: Example * Model | 2969410 |

## F.4 PERFORMANCE DIFFERENCES AMONG IMAGENET SYNSETS

**Table F.2: Differeces between wide and deep architectures on ImageNet synsets with many classes**. Comparison of accuracy of wide (ResNet-50 with $2.8\times$ width in 3rd stage) and deep (ResNet-83) ImageNet models on synsets with $>50$ classes. Note that some synsets are descendants (hyponyms) of others. p-values are computed using a t-test with multiple testing (Holm-Sidak) correction. Results are for the sets of models used to generate blue dots in Figures F.4 and 7. Post-selection effect sizes and testing in the main text use a disjoint set of models.

| Class | # Classes | Wide Acc. | Deep Acc. | Diff. | p-value |
|---|---|---|---|---|---|
| entity | 1000 | $78.0 \pm 0.01$ | $78.0 \pm 0.01$ | $-0.03$ | 0.89 |
| physical entity | 997 | $78.0 \pm 0.01$ | $78.0 \pm 0.01$ | $-0.03$ | 0.89 |
| object | 958 | $78.1 \pm 0.01$ | $78.1 \pm 0.01$ | $-0.04$ | 0.76 |
| whole | 949 | $78.2 \pm 0.02$ | $78.2 \pm 0.01$ | $-0.05$ | 0.48 |
| artifact | 522 | $73.8 \pm 0.02$ | $73.8 \pm 0.02$ | $-0.01$ | 1 |
| living thing | 410 | $83.5 \pm 0.02$ | $83.6 \pm 0.02$ | $-0.10$ | **0.023** |
| organism | 410 | $83.5 \pm 0.02$ | $83.6 \pm 0.02$ | $-0.10$ | **0.023** |
| animal | 398 | $83.3 \pm 0.02$ | $83.4 \pm 0.02$ | $-0.09$ | **0.032** |
| instrumentality | 358 | $73.9 \pm 0.03$ | $74.0 \pm 0.02$ | $-0.02$ | 1 |
| vertebrate | 337 | $83.3 \pm 0.02$ | $83.3 \pm 0.02$ | $-0.08$ | 0.22 |
| chordate | 337 | $83.3 \pm 0.02$ | $83.3 \pm 0.02$ | $-0.08$ | 0.22 |
| mammal | 218 | $82.0 \pm 0.03$ | $82.1 \pm 0.03$ | $-0.09$ | 0.47 |
| placental | 212 | $81.9 \pm 0.03$ | $81.9 \pm 0.03$ | $-0.08$ | 0.66 |
| carnivore | 158 | $81.1 \pm 0.03$ | $81.2 \pm 0.03$ | $-0.09$ | 0.73 |
| device | 130 | $72.9 \pm 0.05$ | $72.9 \pm 0.04$ | $-0.00$ | 1 |
| canine | 130 | $81.3 \pm 0.03$ | $81.4 \pm 0.04$ | $-0.04$ | 1 |
| domestic animal | 123 | $81.0 \pm 0.04$ | $81.0 \pm 0.04$ | $-0.00$ | 1 |
| dog | 118 | $81.6 \pm 0.04$ | $81.6 \pm 0.04$ | $-0.01$ | 1 |
| container | 100 | $72.7 \pm 0.05$ | $72.7 \pm 0.04$ | $0.00$ | 1 |
| covering | 90 | $72.0 \pm 0.05$ | $72.2 \pm 0.05$ | $-0.19$ | 0.13 |
| conveyance | 72 | $83.5 \pm 0.04$ | $83.4 \pm 0.05$ | $0.13$ | 0.65 |
| vehicle | 67 | $83.2 \pm 0.04$ | $83.1 \pm 0.05$ | $0.11$ | 0.76 |
| hunting dog | 63 | $81.2 \pm 0.05$ | $81.2 \pm 0.05$ | $0.01$ | 1 |
| commodity | 63 | $72.2 \pm 0.06$ | $72.6 \pm 0.07$ | $-0.42$ | $\mathbf{5.1 \times 10^{-5}}$ |
| consumer goods | 62 | $72.3 \pm 0.06$ | $72.7 \pm 0.07$ | $-0.41$ | $\mathbf{6.7 \times 10^{-5}}$ |
| invertebrate | 61 | $83.6 \pm 0.05$ | $83.8 \pm 0.04$ | $-0.16$ | 0.37 |
| bird | 59 | $92.5 \pm 0.04$ | $92.7 \pm 0.05$ | $-0.21$ | **0.0018** |
| structure | 58 | $75.9 \pm 0.06$ | $75.5 \pm 0.07$ | $0.42$ | $\mathbf{5.7 \times 10^{-5}}$ |
| matter | 50 | $77.6 \pm 0.05$ | $77.4 \pm 0.05$ | $0.17$ | 0.74 |

