# OpenReview forum: "Do Wide and Deep Networks Learn the Same Things? Uncovering How Neural Network Representations Vary with Width and Depth"
_ICLR.cc/2021/Conference — ICLR 2021 Poster_

### Official Review · AnonReviewer2 · 2020-10-25
**Interesting exploration of how depth and width of deep neural networks affect the learned representations**

**Rating:** 7
**Confidence:** 3

**Review:**

This paper explores core questions related to how depth and width of deep neural networks affect the learned representations. The authors attempt to develop an empirical understanding of the behavior of neural network architectures after training on real-world data. In my humble opinion, the paper is very clearly written, presenting at the beginning of each section the scientific question they try to answer. The questions that the authors try to answer are pertinent, the conclusions are consistent with the results obtained, and statistical tests are employed to check the existence of statistically significant differences between the different approaches tested. I have several questions that the authors should take more as comments or suggestions than criticisms:

- The paper focuses on the family of ResNet models. Do the authors have solid reasons to believe that their findings generalize to other neural models (other ConvNets, recurrent, generative,...) and problems (regression, dense prediction,...?

- Apart from the fact that "some layers that make up the block structure can be removed with minimal performance loss", what are the main practical consequences of the findings of this research?

- The authors state that "on ImageNet, [...] wide networks perform slightly better on classes reflecting scenes, whereas deep networks are slightly more accurate on consumer goods". Do the authors have any intuition about the reason for this behavior? What more general conclusions could be extracted about the performance of deep vs wide neural networks?

- Related to the previous point, in the last years, the recommendation (for instance, in [1]) was to go as deep as possible, and not care too much about wide neural networks. However, if I'm not mistaken, according to the results obtained by the authors "the average accuracy of these groups [deep and wide models] is statistically indistinguishable". Could the authors comment a bit more on this? How all these experimental results corroborate, complement or refute some of the already existing theoretical results regarding depth and width in neural nets?
[1] Eldan, Ronen, and Ohad Shamir. "The power of depth for feedforward neural networks." Conference on learning theory. 2016.

---

> ### Author Response · Authors · 2020-11-20
> **Author response to AnonReviewer2**
>
> We thank the reviewer for the positive feedback and helpful suggestions!
>
> **Whether our findings hold for other neural architectures:** We choose ResNets as the main model class of interest because many common deep neural nets used for image classification adopt the idea of residual connections to a certain extent (see [1, 3, 4]).
> However, we have also added CKA results on a different architecture (see Appendix C) that contains the same set of layers as ResNets but has no residual connections. In this case, we also observe the block structure phenomenon in larger capacity models. This shows that our findings should generalize to standard convolutional architectures.
> Regarding the class of problems, we choose to focus on image classification in this work to make the investigation tractable. We believe our findings should generalize to other domains and tasks, but we leave these experiments for future work.
>
> **Intuition about the differences in class performance between wide and deep networks:** We hypothesize that the differences between wide and deep networks on scenes and consumer goods reflect the way the receptive fields evolve through the network. When smaller details are important to the task, it might be advantageous to have wider layers, whereas when global structure is important, depth may be more useful. However, we have not performed rigorous experiments to test this hypothesis, and we leave this question to future work.
>
> **Recommendation for going wide or deep:** Comparisons between wide and deep networks in Section 7 are intentionally demonstrated for architectures with comparable test accuracies, so as to remove performance as a confounder for representational structure. Regarding the best practices, our literature review on network design so far suggests that it’s important to balance both depth and width to achieve optimal performance (see [1,2]).
>
> [1] Tan, M., & Le, Q. (2019, May). EfficientNet: Rethinking Model Scaling for Convolutional Neural Networks. In International Conference on Machine Learning (pp. 6105-6114). http://proceedings.mlr.press/v97/tan19a/tan19a.pdf
>
> [2] Zagoruyko, S., & Komodakis, N. (2016). Wide residual networks. BMVC. https://arxiv.org/pdf/1605.07146.pdf
>
> [3] Howard, Andrew G., et al. "Mobilenets: Efficient convolutional neural networks for mobile vision applications." arXiv preprint arXiv:1704.04861 (2017).
> https://arxiv.org/pdf/1704.04861.pdf
>
> [4] He, Kaiming, et al. "Mask r-cnn." Proceedings of the IEEE international conference on computer vision. 2017.
> https://arxiv.org/pdf/1703.06870.pdf

---

### Official Review · AnonReviewer1 · 2020-10-28
**engaging approach that opens more questions**

**Rating:** 6
**Confidence:** 3

**Review:**

This paper explores characteristics of resnet networks that arise with different capacity.  In particular, a blockwise structure of behavioral similarity arises when the network has large capacity, regardless of whether that capacity was introduced with greater depth or with greater width (similarity is measured using CKA, which compares the pattern of pairwise comparisons among elements in a minibatch between two different sets of layers/activations).

This is an interesting method and characterization of resnet behavior, with thorough experiments that tie together different aspects of the approach:  CKA is used to show a type of blockwise similarity, much of which is subsequently explained, and related experimentally to classification performance using linear probes through the layers.  Finally, it opens questions for future research, such as: despite qualitatively similar layerwise behavior, why does larger depth vs larger width produce different types of classification errors?  Might any of the differences in patterns that develop for deep vs wide in CKA relate to the spatial scale of regions that most influence each layer (the effective receptive fields)?

Overall this looks like a rather thorough investigation.  I have a few questions of things I don't quite know how to interpret, below, which I feel could be made clearer.



Questions:

- Fig 1:  In the last two plots of each of the top two rows, it appears the block starts to develop from the "middle" out, in two sections, as opposed to from the first or last layer in a section.  Is there a reason for this and what does it correspond to?

- Sec 5.1:  Is there an intuitive description of the first principal component in R^n, as opposed to R^p1 --- it seems easier to think of the latter, being the activation vector that captures the most variance over n datapoints, but the one used here is from the transpose, with the vector in R^n (i.e. coefficients of data samples):  What does this mean and how does it relate to the block structure and/or residual connections?
- Given that much of the blockwise structure is due to the first principal component, can anything be said about the structures that arise after removing it?  How do these behave when increasing depth vs width and do these relate at all to the different classification results?

---

> ### Author Response · Authors · 2020-11-20
> **Author response to AnonReviewer1**
>
> Thank you for the helpful feedback and comments!
>
> **Block develops from “middle” out:** a possible heuristic explanation is that the blocks develop “away” from layers that perform spatial downsampling, which happen at the beginning of different sections of the network. However, there are variations to the block size and position across initializations (Figure 6), and there remain interesting open questions on the dynamics and frequency of different types of block structure.
>
> **Intuitive description of the first PC in R^n:** This is the vector that best reconstructs the similarity between data points. We perform this analysis because it is a logical thing to do given the decomposition of CKA in terms of PCs, and because it provides an alternative way to think about what CKA is measuring.
>
> **Effect of removing first PC on block structure:** In the submitted version, we studied the effect of removing the first PC on networks with the block structure in Figure 3, finding that removing the first PC highly reduces the block structure. In the revised version, we have included the full picture of how the representational structure varies with increasing depth and width after the first PC is removed (Appendix Figure D.6). We find that removing the first PC has little effect on smaller models (with no block structure), since the first PC only accounts for a small fraction of the variance in representations in these models.

---

### Official Review · AnonReviewer4 · 2020-10-28
**Good paper but the choice of similarity function is not clear**

**Rating:** 8
**Confidence:** 5

**Review:**

The paper aims to get a better understanding of differences between wide and deep neural networks through an empirical evaluation. It does it through a similarity analysis, and an analysis of which errors wide and deep residual networks do.

The paper is well written and should not be difficult to reproduce. Most of the questions I got while reading where addressed in the paper, in an extensive evaluation. The most interesting and somewhat surprising finding is that even though two networks with different number of parameters and layers but with the same accuracy make very different mistakes, and there is a pattern to it. The weakest part is the similarity analysis, which does not seem to reveal much new. It has already been known that deep networks have layers which do not contribute significantly to final accuracy, can be proved or even forced to learn more useful representation. Would be interesting to apply the similarity analysis in a network with reinitialized layers. Also, the choice of specific similarity function and it's benefits/drawbacks are not discussed. Could the same analysis be achieved with a simpler similarity function? Is it computationally efficient to compute?

Overall, there are definitely valuable contributions in the paper, so I propose lower score only due to the unclear choice of similarity function, as described above.

---

> ### Author Response · Authors · 2020-11-20
> **Author response to AnonReviewer4**
>
> Thank you for your time and feedback! We are very happy to hear that you enjoyed the paper!
>
> **Similarity analysis and CKA:** In the submitted version of the paper, due to space constraints, we referred the reader to [1] for further explanation of the rationale and the details of the similarity technique (CKA). In response to your feedback, we have now included this information in the main text of the updated paper (Section 3.1) and additional details in Appendix A. We repeat this description below.
>
> We use linear centered kernel alignment (CKA) (Kornblith et al., 2019; Cortes et al., 2012) to measure similarity between neural network hidden representations. Let $\mathbf{X} \in \mathbb{R}^{m \times p_1}$ and $\mathbf{Y} \in \mathbb{R}^{m \times p_2}$ contain activations of two layers, one with $p_1$ neurons and another $p_2$ neurons, to the same set of $m$ examples. The elements of the Gram matrices $\mathbf{K} = \mathbf{X}\mathbf{X}^\mathsf{T}$ and $\mathbf{L} = \mathbf{Y}\mathbf{Y}^\mathsf{T}$ reflect the similarities between pairs of examples according to the representations contained in $\mathbf{X}$ and $\mathbf{Y}$. Let $\mathbf{H} = \mathbf{I}_n - \frac{1}{n}\mathbf{1}\mathbf{1}^\mathsf{T}$ be the centering matrix. Then $\mathbf{K}' = \mathbf{H}\mathbf{K}\mathbf{H}$ and $\mathbf{L}' = \mathbf{H}\mathbf{L}\mathbf{H}$ reflect the similarity matrices with their column and row means subtracted. HSIC measures the similarity of these reshaped similarity matrices by reshaping them to vectors and taking the dot product between these vectors, $\mathrm{HSIC}_0(\mathbf{K},\mathbf{L}) = \mathrm{vec}(\mathbf{K}') \cdot \mathrm{vec}(\mathbf{L}')/(m-1)^2$. HSIC is invariant to orthogonal transformations of the representations and, by extension, to permutation of neurons, but it is not invariant to isotropic scaling of the original representations. CKA further normalizes HSIC to produce a similarity index between 0 and 1 that is invariant to isotropic scaling,
> \begin{align}
>     \mathrm{CKA}(\mathbf{K}, \mathbf{L}) = \frac{\mathrm{HSIC}_0(\mathbf{K}, \mathbf{L})}{\sqrt{\mathrm{HSIC}_0(\mathbf{K}, \mathbf{K}) \mathrm{HSIC}_0(\mathbf{L}, \mathbf{L})}}.
> \end{align}
> Kornblith et al. (2019) have shown that, when measured between layers of networks trained from different random initializations, CKA can reliably identify corresponding layers in architecturally identical networks trained from different initializations, whereas other previously proposed ways of measuring similarity of neural network representations do not.
>
> **Deep networks have layers which do not contribute significantly to final accuracy:** While it is not surprising that a deep network may have layers with limited contribution to final accuracy on its own, successive layers may still progressively refine the network’s representation. More specifically, network depth alone does not imply the existence of the block structure (a large set of contiguous layers which precisely function by preserving and propagating the first principal component). Indeed, the ability to (quantitatively) analyze representations in sets of hidden layers has precisely been hindered by a lack of techniques to understand hidden representation similarity in neural networks, which is made complex due to the size, complexity and distributed nature of these representations. We are able to utilize CKA (which is also shown in [1] to be more reliable compared to other methods that measure representation similarity) to address these issues and provide new insights into groups of layer representations.
>
> **Reinitialized layers:** we are not sure what exactly the reviewer means by this, but we think it refers to random initialization. As part of future work, we are currently studying the learning dynamics of the block structure.
>
> **Choice of similarity function:** the justification for the choice of similarity function has now been added to the revised version (see earlier point). It is very efficient to compute, particularly in its minibatch form which we state and prove in the paper (Section 3.1 and Appendix Section A). In addition, the submission also included a complementary analysis of block structure using cosine similarity (Figure 3). Cosine similarity cannot be used in general because hidden representations in different parts of the network have different dimensionalities, and it is also sensitive to rotations in activation space that do not affect what information can be extracted from the representation by subsequent layers. However, we can apply it to just the first principal component of each representation, which is aligned with the function of the block structure as preserving and propagating this first principal component.
>
> [1] Kornblith, S., Norouzi, M., Lee, H., & Hinton, G. (2019, May). Similarity of Neural Network Representations Revisited. In International Conference on Machine Learning (pp. 3519-3529). http://proceedings.mlr.press/v97/kornblith19a/kornblith19a.pdf

---

### Official Review · AnonReviewer3 · 2020-10-30
**This paper studies the effects of width and depth on neural network representation.**

**Rating:** 6
**Confidence:** 3

**Review:**

In this paper，the author studies the effects of width and depth on neural network representation.


In this paper，the author studies the effects of width and depth on neural network representation. This paper conducts lots of experiments on CIFAR-10, CIFAR-100 and ImageNet with different network architectures and apply the CKA to measure the similarity between representation of each layer. As a result, they find a characteristic block structure in the hidden representations of larger capacity models which is also dependent to the size of dataset. This work has the following advantages:
1、	Well-arranged and detailed experiments which strongly support the final conclusion.
2、	Exquisite figures that well displays the experiment results.
3、	The concept of “block structure” in ResNet is novel. And all of the experiments and analysis illustrate there really exists blocks with similar representation in overoptimization models. And this phenomenon can guide researchers to design networks well.
However, there are some disadvantages or doubts in my opinion:
1、	This paper lacks of further explanations about the CKA or HSIC tools. I can’t fully understand how the similarities between representations of each layer are measured.
2、I wonder if the block structure arises dependent to the residual blocks. I want to see more experiments with other network architectures.
3、I think the analysis of effects of width and depth on neural network representation can well guide researchers to design networks. So，I expect to see an modified network architecture or a method to balance the network size and accuracy . However, this paper is just about theoretical analysis based on experiment phenomenon. Thus, I think this paper is lack of some innovation.

4. Some previous related works is better to be appreciated:

C. L. Philip Chen et al Broad Learning System: An Effective and Efficient Incremental Learning System Without the Need for Deep Architecture，IEEE Transactions on Neural Networks and Learning Systems，2017
C.L.P. Chen, Z. Liu, and S. Feng, Universal approximation capability of broad learning system and its structural variations， IEEE transactions on neural networks and learning systems 30 (4), pp. 1191-1204.

---

> ### Author Response · Authors · 2020-11-20
> **Author response to AnonReviewer3**
>
> We thank the reviewer for their comments and feedback!
>
> **Further explanation of CKA/HSIC:** In the original submitted paper, due to space constraints, we referred readers to [1] for details regarding CKA and further validation of its utility for measuring similarity of neural network representations. But informed by your feedback, we have now included more in-depth descriptions of what CKA measures in Section 3.1.
>
> At a high level: neural network hidden representations are challenging to analyze for several reasons including (i) their large size, (ii) their distributed nature, where important features in a layer may rely on multiple neurons, and (iii) lack of alignment between neurons in different layers. Centered kernel alignment (Kornblith et al., 2019; Cortes et al., 2012) addresses these challenges, providing a robust way to quantitatively study neural network representations, specifically, to compute similarity of representations. As shown in Kornblith et al. (2019) CKA is a highly robust quantitative measure, e.g. being able to reliably identify corresponding layers in architecturally identical networks trained from different initializations, whereas other previously proposed ways of measuring similarity of neural network representations do not.
>
> We have expanded the mathematical details of CKA in Section 3.1, which we have also copied below:
> Let $\mathbf{X} \in \mathbb{R}^{m \times p_1}$ and $\mathbf{Y} \in \mathbb{R}^{m \times p_2}$ contain activations of two layers, one with $p_1$ neurons and another $p_2$ neurons, to the same set of $m$ examples. The elements of the Gram matrices $\mathbf{K} = \mathbf{X}\mathbf{X}^\mathsf{T}$ and $\mathbf{L} = \mathbf{Y}\mathbf{Y}^\mathsf{T}$ reflect the similarities between pairs of examples according to the representations contained in $\mathbf{X}$ and $\mathbf{Y}$. Let $\mathbf{H} = \mathbf{I}_n - \frac{1}{n}\mathbf{1}\mathbf{1}^\mathsf{T}$ be the centering matrix. Then $\mathbf{K}' = \mathbf{H}\mathbf{K}\mathbf{H}$ and $\mathbf{L}' = \mathbf{H}\mathbf{L}\mathbf{H}$ reflect the similarity matrices with their column and row means subtracted. HSIC measures the similarity of these reshaped similarity matrices by reshaping them to vectors and taking the dot product between these vectors, $\mathrm{HSIC}_0(\mathbf{K},\mathbf{L}) = \mathrm{vec}(\mathbf{K}') \cdot \mathrm{vec}(\mathbf{L}')/(m-1)^2$. HSIC is invariant to orthogonal transformations of the representations and, by extension, to permutation of neurons, but it is not invariant to isotropic scaling of the original representations. CKA further normalizes HSIC to produce a similarity index between 0 and 1 that is invariant to isotropic scaling,
> \begin{align}
>     \mathrm{CKA}(\mathbf{K}, \mathbf{L}) = \frac{\mathrm{HSIC}_0(\mathbf{K}, \mathbf{L})}{\sqrt{\mathrm{HSIC}_0(\mathbf{K}, \mathbf{K}) \mathrm{HSIC}_0(\mathbf{L}, \mathbf{L})}}.
> \end{align}
>
> **Effect of residual blocks:** In Appendix C we have added additional results on a simple variant of ResNets, which contains the same constituent layers but no residual connections. This architecture thus resembles a standard convolutional network. Since the lack of residual connections prevents very deep networks from performing well on the task, we only plot the representational similarity for models of increasing width. This new set of results shows that the block structure phenomenon does not depend on the presence of residual blocks.
>
> **Guiding researchers to design networks:** The main focus of this work was to study how representations varied with changing width and depth, which provided numerous novel findings. We discovered the block structure in the process, which we studied extensively, and identified connections to training dataset size, its precise function on the representation principal components and methods to collapse the block structure. This also led to a better understanding of representation similarity across different architectures and motivated our study of characteristic errors. Designing a new class of networks to prevent the emergence of block structures (and thus redundancy in representations) is unfortunately beyond the scope of this work, but we hope our findings provide an important foundation for future research on principled exploration of neural network architectures.
>
> **Additional References:** Thank you for the additional references, we are reading them through!
>
> [1] Kornblith, S., Norouzi, M., Lee, H., & Hinton, G. (2019, May). Similarity of Neural Network Representations Revisited. In International Conference on Machine Learning (pp. 3519-3529). http://proceedings.mlr.press/v97/kornblith19a/kornblith19a.pdf

---

### Decision · Program_Chairs · 2021-01-07
**Final Decision**

**Decision:**

Accept (Poster)

**Comment:**

This paper studies whether neural networks with different architectures, especially different width and depth, learn similar representations. All reviewers agree that the investigations are thorough and the experimental discoveries are convincing and well explained. Good work. I recommend accept.